# ClpC1-targeting peptide natural products differentially dysregulate the proteome of *Mycobacterium tuberculosis*

Isabel K. Barter [1,2], Max J. Bedding[1,2], Julia Leodolter [3], Joshua W. C. Maxwell[1,2], Paige M. E. Hawkins[1,2], Maxwell T. Stevens [4], Matthew B. McNeil [5], William J. Jowsey[5], Trixie Wang[4], Diana Quan[4], Sabryna Junker [3], Manuela Flórido[6], Daniel Hesselson[6], Gregory M. Cook [5], Tim Clausen [3], Warwick J. Britton [4,7] ✉, Mark Larance [8] ✉ & Richard J. Payne [1,2] ✉

Targeting the protein quality control system in *Mycobacterium tuberculosis* represents a promising and underexplored opportunity for antibiotic development. The ClpC1:ClpP1P2 protease is an essential component of the system that mediates both regulatory and stress-related protein degradation. Several non-ribosomal peptide natural products, including ecumicin, ilamycins (rufomycins) and cyclomarins, have been discovered that bind to the ClpC1 chaperone of the complex and exhibit potent antimycobacterial activity, leading to significant interest in the ClpC1:ClpP1P2 system as a bona fide target for the new tuberculosis drugs. In this study, we combine quantitative proteomics, bioinformatics, transcriptomics, CRISPRi knockdown, and targeted biochemical and biophysical assays to dissect the mechanisms of ecumicin, ilamycin and cyclomarin in clinically relevant *Mycobacterium tuberculosis*. Strikingly, despite exhibiting similar binding modes to ClpC1, each compound induces distinct effects on protein degradation. Notably, ilamycin and ecumicin do not trigger the ClpC2 rescue mechanism that mitigates cyclomarin-induced mycobacterial toxicity. In addition, we identify a novel interaction between ecumicin and stress-response chaperone Hsp20. The differential disruption of ClpC1 substrates, stress-response chaperones, and distinct reshaping of the *Mycobacterium tuberculosis* proteome by the three natural products, unveils new opportunities for the development of protein quality control-targeted antimycobacterials.

Tuberculosis (TB), caused by *Mycobacterium tuberculosis* (*Mtb*), is one of the leading causes of death worldwide[1]. The bacteria are transmitted via airborne droplets that typically infect the lungs and cause structural damage to the organ[2,3]. It is estimated that a quarter of the global population is infected with latent *Mtb* and are therefore at risk of developing active disease. In 2023, the World Health Organisation reported that TB was responsible for an estimated 1.25 million deaths, with an additional 10.8 million morbidities[1]. Current TB antibiotics generally target a small number of common biological pathways; cell wall synthesis, ATP synthesis, transcription, and translation[4,5]. The

overuse and misuse of frontline antibiotics has contributed to the steady incidence of multi-drug resistant TB and extensively-drug resistant TB, which poses a significant threat to the future control of the disease and therefore to global health[1,6]. It is widely accepted that there is a crucial need for the development of new antimycobacterials that target novel pathways, are fast acting, and non-toxic to reduce the TB disease burden globally[1].

Proteostasis is guided by environmental signals and executed through the balance of synthesis, folding and degradation[7,8]. The capacity of a pathogen to detect environmental cues and adapt accordingly is a key determinant of its virulence[9]. Several first- and second-line TB drugs used clinically work by disrupting biosynthetic machinery at various stages of protein production, including rifampicin, streptomycin, capreomycin, amikacin, and linezolid[10,11]. Beyond protein synthesis, perturbation of protein quality control (PQC) is a comparatively unexplored approach for the generation of potent anti-TB agents. Chaperones and proteases are the key components of the PQC system, responsible for the maintenance of native protein conformations and clearing terminally damaged proteins[8,12]. Through binding to misfolded proteins, chaperones minimise the consequences of denaturation and prevent the formation of intracellular aggregates. In the event a protein becomes irreparably damaged, it is degraded by proteases enabling amino acid building blocks to be recycled.

*Mtb* contains several proteases, including Clp, FtsH, and eukaryotic-like proteases[9]. A functional Clp proteolytic complex is essential for growth and survival of *Mtb*, coordinating post-translational regulation and proteome maintenance[13–15]. The *Mtb* Clp protease (ClpP1P2) is comprised of two heptameric serine proteases, ClpP1 and ClpP2, that associate to form an active cylindrical chamber containing 14 active sites for hydrolysis of peptide bonds within proteins[16–18]. Regulated proteolysis requires the association of ATPase chaperones, such as ClpC1, to form the ClpC1:ClpP1P2 (ClpC1P1P2) complex (Fig. 1a). Active ClpC1 exists as a hexamer that constitutes a central axial pore for substrate interactions. Each ClpC1 monomer contains an N-terminal domain (NTD) connected via an inherently disordered 26 amino acid linker to the ATPase domains, D1 and D2[19,20]. The D1 and D2 domains hydrolyse ATP to actively unfold and translocate client proteins to the proteolytic core of ClpP1P2 for degradation[18,19]. Substrate recognition by *Mtb* ClpC1 is complex and not well defined. However, recent advances have demonstrated that recognition is influenced by features such as unstructured N- or C- terminal regions, which facilitate engagement with ClpC1-NTDs[14,21].

The relevance of the Clp system as a potential antimycobacterial target was reinforced by the discovery of several ClpC1-binding peptide natural product antibiotics, including non-ribosomal peptides (NRPs) ecumicin, cyclomarins, ilamycins (also called rufomycins) and the ribosomally-synthesised lassomycin[22–25]. In this study, we focus on the three main NRP classes. Ecumicin is a non-ribosomal trideca-depsipeptide isolated from the actinomycete *Nomonurea* sp MJM5123. The molecule contains extensive backbone *N*-methylation and two non-canonical amino acids: *N*-methyl-4-methoxy-L-tryptophan and *threo*-β-hydroxy-L-phenylalanine[22,26]. The ilamycins/rufomycins and cyclomarins are both cyclic heptapeptides produced by *Streptomyces* sp[23,24]. Ilamycin E₂ (Ila E), the most potent congener in its family, contains several non-canonical amino acids, including: L-2-amino-4-hexenoic acid, *N*-tert-prenylated L-tryptophan, L-3-nitrotyrosine, and an interesting hydroxypiperidinone motif[23]. Cyclomarin C contains *N*-tert-prenylated β-hydroxytryptophan, 2-amino-3,5-dimethyl-4-hexenoic acid, 5-hydroxyleucine, and β-methoxyphenylalanine[27]. Each NRP has been shown to bind to the NTD of ClpC1 and proposed to mimic exposed hydrophobic regions of unfolded proteins that would normally be cleared through ClpC1:ClpP1P2 proteolysis[19,28–30]. There is considerable overlap in the ClpC1-binding interface between the three

compounds and mutagenesis studies have revealed mutations at F80 and V13 confer resistance to more than one natural product and a shared contact, K85, influences degradation capacity in the presence of cyclomarin (Fig. 1b and Supplementary Fig. 1a)[28–31]. Distinct effects on the ATPase activity and degradation of model substrates have also been reported[30,32]. Taken together, this suggests an elaborate and nuanced interplay between substrate recognition and NRP activity, yet the broader effects on the *Mtb* proteome and native substrates are yet to be elucidated. Leveraging their capacity to bind to the ClpC1-NTD, one of these NRPs, cyclomarin, recently inspired the design of bacterial proteolysis targeting chimeras (BacPROTACS), enabling targeted protein degradation in mycobacteria[33–35]. Given the emerging interest in targeting proteolysis in *Mtb*, furthering our understanding of the underlying mechanisms of the each of these NRPs is crucial for informing development of novel and innovative Clp-targeting antibacterials in the future.

In this study, we combined systematic proteomics, bioinformatics, transcriptomics, CRISPRi knockdown, and in vitro biochemical assays to investigate the effects of three NRPs, an ecumicin analogue (Ecu*)[36], ilamycin E₂ (also termed rufomycin 22)[37], and desoxycyclomarin (dCym)[34] on clinically relevant *Mtb* H37Rv. Each compound caused distinct, substrate specific dysregulation of ClpC1. Small stress response chaperones were differentially affected. All three NRPs bound ClpC2, but only dCym triggered its upregulation and exhibited enhanced activity in ClpC2 knockdown strains. Hsp20 was strongly upregulated by Ecu* and, to a lesser extent IlaE, with binding assays confirming a direct Ecu*: Hsp20 interaction. Collectively, these results demonstrate that the NRPs perturb PQC through distinct mechanisms, including differential dysregulation of ClpC1 substrate recognition and engagement of stress-response chaperones.

## Results

### NRPs cause widespread changes to the *Mtb* proteome
In this study, we have examined the activity of a representative congener (or related synthetic analogue) from the three families of ClpC1-binding NRPs, including: a synthetic ecumicin analogue (Ecu* or **1**) with a simplified amino acid composition, and superior antimycobacterial potency, to the parent natural product (previously described by Hawkins et al.)[36], ilamycin E (IlaE or **2**)[38], and a simplified synthetic analogue of cyclomarin A, desoxycyclomarin (dCym or **3**)[39] (Fig. 1c and Supplementary Fig. 1b). Each compound exhibited potent antimycobacterial activity against virulent *Mtb* strain H37Rv with these data used to direct downstream experiments (Fig. 1c and Supplementary Fig. 1c).

The NRPs target the chaperone component of the ClpC1P1P2 proteolysis system[28–30,34,40]. To assess their impact on the *Mtb* proteome, we employed label-free quantification (LFQ) proteomics to monitor changes in protein abundance. Sublethal doses and treatment durations were established for each NRP derivative (**1–3**) to minimise cell death, which could confound proteomic analysis. Initial assessment of (**1**) at 16 h, below one doubling time, revealed minimal changes (Supplementary Fig. 2a). Pronounced, reproducible effects emerged after 48 h, which was selected for all subsequent analysis maintaining 70–80% mycobacterial survival (Supplementary Fig. 2b, c). Stability of each NRP in culture media for 48 h was confirmed by LC-MS (Supplementary Fig. 2d). Following NRP exposure, cells were lysed using SDS with an overnight freezing step and ultra sonication to maximise extraction efficiency. Proteins were then trypsin-digested, peptides desalted and analysed by quantitative LC-MS/MS (Fig. 1d). We detected 3175 proteins across all samples, achieving ~80% coverage of the known *Mtb* proteome (Supplementary Data 1). Hundreds of proteins were significantly altered after NRP treatment (Benjamini–Hochberg-adjusted $p \leq 0.05$, $\log_2$FC > 0.5 or < −0.5, Fig. 1e−g and Supplementary Data 1). Ecu* (**1**) caused the most widespread effects, with nearly 17% of the proteome altered and 36 proteins showing changes between ten

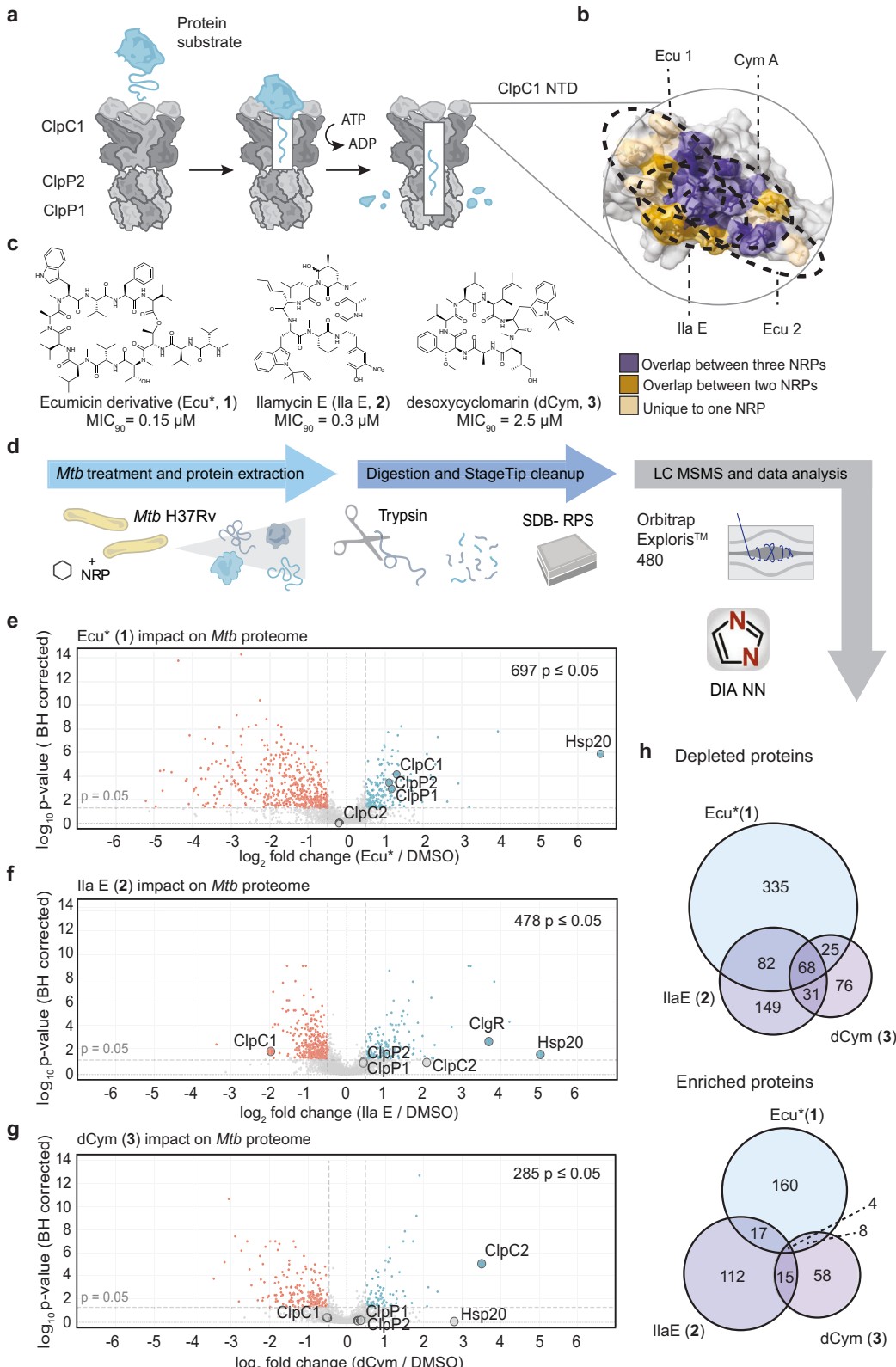

**Nature Communications** | (2026)17:1725

and 96-fold. IlaE (**2**) and dCym (**3**) affected 12% and 7% of the proteome, respectively with fewer large changes (Fig. 1e, f). These data provide evidence that NRPs alter a substantial portion of the 4173 protein *Mtb* proteome[41] through ClpC1 dysregulation.

Although all three NRPs bind the ClpC1 NTD, their proteomic consequences were distinct. The changes induced by IlaE (**2**) showed greater overlap with those caused by Ecu* (**1**) and dCym (**3**) than Ecu* (**1**) and dCym (**3**) did with each other, suggesting that Ecu* (**1**) and dCym (**3**) elicit the most divergent responses. Only 72 proteins were consistently affected across all treatments and most of these were depleted (68 of 72, Fig. 1h), suggesting each NRP alters ClpC1 substrate recognition in different ways.

**Fig. 1 | Proteomics reveals widespread changes to *Mtb* H37Rv proteome following treatment with NRPs. a** Schematic representing protein degradation facilitated by ClpC1P1P2. ClpC1 recognises, unfolds and translocates substrates to the ClpP1P2 components for proteolysis. **b** NRP ClpC1 binding surfaces overlap. Residues with ≥ 40 Å Van der Waals overlap were considered to contribute to the NRP:ClpC1 interface (PDB 6PBS, PDB 6CN8, PDB 3WDC). Residues which were in close proximity to all three compounds are coloured purple, those that are close to two compounds coloured brown, and those close to only one compound coloured light brown. **c** Structures of natural product derivatives analysed in this study; Ecumicin derivative (Ecu*, **1**), Ilamycin E (IlaE, **2**), desoxycyclomarin (dCym, **3**). Minimum inhibitory concentration (MIC) to sterilise 90% of bacteria (MIC$_{90}$) indicated below each compound. Raw data from the resazurin MIC assay found in Supplementary Fig. 1c and source data are provided in Source data file. **d** Schematic of *Mtb* treatment, protein extraction and sample preparation workflow for mass spectrometry-based proteomics. **e–g** Volcano plots representing the effects of the NRPs on the proteome of *Mtb* H37Rv (**e** = Ecu*, **f** = IlaE and **g** = dCym). Statistical significance was calculated with a one-way ANOVA, and *p*-values were BH-adjusted. Red dots represent depleted proteins (<−0.5 log$_2$ fold change, *p*-value < 0.05). Blue dots represent enriched proteins ( > 0.5 log$_2$ fold change, *p*-value < 0.05). Clp proteins and highly upregulated Hsp20 are labelled. DMSO *n* = 7, Ecu* (**1**) *n* = 6, IlaE (**2**) *n* = 4, dCym (**3**), *n* = 4. **h** Venn diagrams showing number of similarly or uniquely affected proteins, enriched (top) or depleted (bottom) following NRP treatment. Source data provided in Supplementary Data 1.

To probe impacted pathways, gene ontology enrichment revealed broadly similar clusters across compounds, despite differences in individual proteins (Supplementary Fig. 3 and Supplementary Data 2). Depleted processes included stress responses and diverse metabolic pathways (lipid, amino acid, small molecule and macromolecule metabolism). Ecu* (**1**) additionally depleted transcription-related proteins while enriching those linked to fatty acid synthesis and stress response processes. IlaE (**2**) and dCym (**3**) primarily depleted fatty-acid biosynthesis proteins, with few enriched pathways. These data are consistent with ClpC1 interaction studies[14,42], reinforcing the widespread role of the ClpC1P1P2 system in essential mycobacterial processes.

### BTZ broadly inhibits protein degradation in *Mtb*
Bortezomib (BTZ) is a well-characterised eukaryotic proteosome inhibitor that also dysregulates the ClpP1P2 protease complex (Fig. 2a, b)[43–45]. BTZ inhibited *Mtb* growth with a MIC$_{50}$ of 5.6 μM (Supplementary Fig. 4a), consistent with previous reports and both 16 h and 48 h incubations resulted in sublethal conditions (Supplementary Fig. 4b). Using LFQ proteomics, we observed 325 and 881 differentially abundant proteins at 16 and 48 h-post treatment, respectively. More proteins were enriched than depleted at both timepoints, reflecting impaired protein turnover owing to ClpP1P2 and *Mtb* proteasomal inhibition (Fig. 2c and Supplementary Data 4). Notably, the number of depleted proteins increased at 48 h, consistent with prolonged inhibition of proteolysis suppressing protein synthesis and transcription[46]. While recent in vitro studies show BTZ can activate ClpC1P1P2 model substrate proteolysis[45], our data demonstrate net inhibition of protein degradation within *Mtb* cells.

Comparisons of BTZ-induced changes with those elicited by Ecu*(**1**), IlaE (**2**), and dCym (**3**) revealed limited overall overlap, with IlaE (**2**) and Ecu* (**1**) showing greater similarity to BTZ than dCym (**3**) (Fig. 2d, e). ClpC1P1P2 substrates comprise regulatory targets and terminally damaged proteins that are generated through stress and translational errors. Validated regulatory ClpC1P1P2 substrates include PanD, Hsp20, ClgR, CarD, and Vap/Rel antitoxins[14,15,42,47,48]. In addition to ClpP1P2, BTZ also inhibits the 20S eukaryotic-like protease in *Mtb*[49]. The 20S proteosome recognises substrates tagged with the prokaryotic ubiquitin-like protein, Pup, which serves as a degradation signal. Known 20S proteosome substrates include Mpa, HspR, FabD, and Pup itself[50–52]. The NRPs did not trigger global enrichment of all ClpC1P1P2 substrates, instead they exhibited substrate-dependent modulation. Ecu* (**1**) significantly enriched PanD and Hsp20 while depleting CarD, and IlaE (**2**) enriched ClgR and Hsp20 (Supplementary Fig. 5a). As anticipated, the NRPs did not lead to the enrichment of 20S proteosome substrates (Supplementary Fig. 5b). Several Vap/Rel antitoxins were differentially affected, being either enriched or depleted by Ecu*(**1**) and IlaE (**2**) (Supplementary Fig. 5c). Interestingly, ClgR and Hsp20 were the two most enriched proteins in response to BTZ (**4**) (Fig. 2c). Both were also strongly enriched by IlaE (**2**), whereas only Hsp20 was enriched by Ecu* (**1**) and neither were enriched by dCym (**3**) (Fig. 1e–g and Supplementary Fig. 5a). BTZ broadly enriched both ClpC1P1P2 and 20S proteosome substrates, confirming a general inhibitory impact on protein degradation within *Mtb* cells (Fig. 2f, g and Supplementary Fig. 5d). Together, these data indicate that, unlike BTZ, the NRPs modulate ClpC1 chaperone activity rather than global inhibition of mycobacterial protease activity.

### Intrinsic protein disorder and substrate-specific effects of NRPs
ClpC1P1P2 also clears unfolded proteins to prevent aggregation[18,53–55] (Fig. 3a). We examined the relationship between NRPs and intrinsically disordered proteins that contain regions of disorder, resembling unfolded polypeptides, yet are functional and stable rather than aggregation-prone[56]. Using the mobiDB server[57,58] AlphaFold2-derived disorder predictions, which cover 68% of the *Mtb* proteome, *Mtb* proteins were assigned disorder scores. Statistical analysis revealed that intrinsically disordered proteins were 5.68 times more likely to be depleted with Ecu* (**1**), yet 4.76 and 6.25 times more likely to be enriched with IlaE (**2**) and dCym (**3**), respectively (Fig. 3b and Supplementary Table 1). In contrast, proteins with disordered termini did not show statistically significant global enrichment or depletion in the context of the NRPs (Fig. 3c, d, Supplementary Table 2, and Supplementary Data 3). These results indicate that overall intrinsic disorder, rather than terminal disorder, is more predictive of susceptibility to NRP-induced changes. However, as this conclusion is derived from a global proteomic analysis, it likely encompasses both direct, ClpC1 dysregulation-dependent effects, and broader protein responses that would result from the dramatic changes to the *Mtb* proteome that occurs following NRP treatment. Because these effects cannot be readily disentangled, genuine associations with terminal disorder may be underrepresented in the aggregated data.

### Perturbation of ClpC1P1P2 substrate degradation by NRPs
Proteomic analysis captures protein abundance at a specific point in time but does not directly measure turnover. To complement our observations, we performed in vitro degradation assays using both a model disordered protein and physiologically relevant ClpC1P1P2 substrates. Degradation of β-casein, the archetypal disordered protein model[34], was strongly inhibited by Ecu* (**1**), whereas IlaE (**2**) and dCym (**3**) had no observable effect (Fig. 3d and Supplementary Fig. 6a). It is worth noting that BTZ (**4**) has been described as an activator of the *Mtb* ClpC1P1P2 protease in vitro, particularly at low concentrations[45]. We did not observe a discernible effect on β-casein degradation in the presence of BTZ, which likely reflects the relatively high concentration utilised in the assay and the absence of a fluorescence-based reporter, which limits assay sensitivity (Supplementary Fig. 6a).

We next examined PanD and Hsp20, two native *Mtb* substrates with disordered C-terminal regions and demonstrate in vitro evidence of their degradation by ClpC1P1P2 and the influence of the NRPs on this degradation. In contrast to β-casein, each of the NRPs inhibited PanD degradation in vitro, despite only Ecu* (**1**) triggering an enrichment of PanD in *Mtb* cells in our proteomics experiments (Fig. 3e and Supplementary Fig. 6b). Hsp20 degradation was inhibited only by dCym (**3**). Despite strong enrichment of Hsp20 by Ecu*(**1**), IlaE (**2**) and BTZ (**4**) in

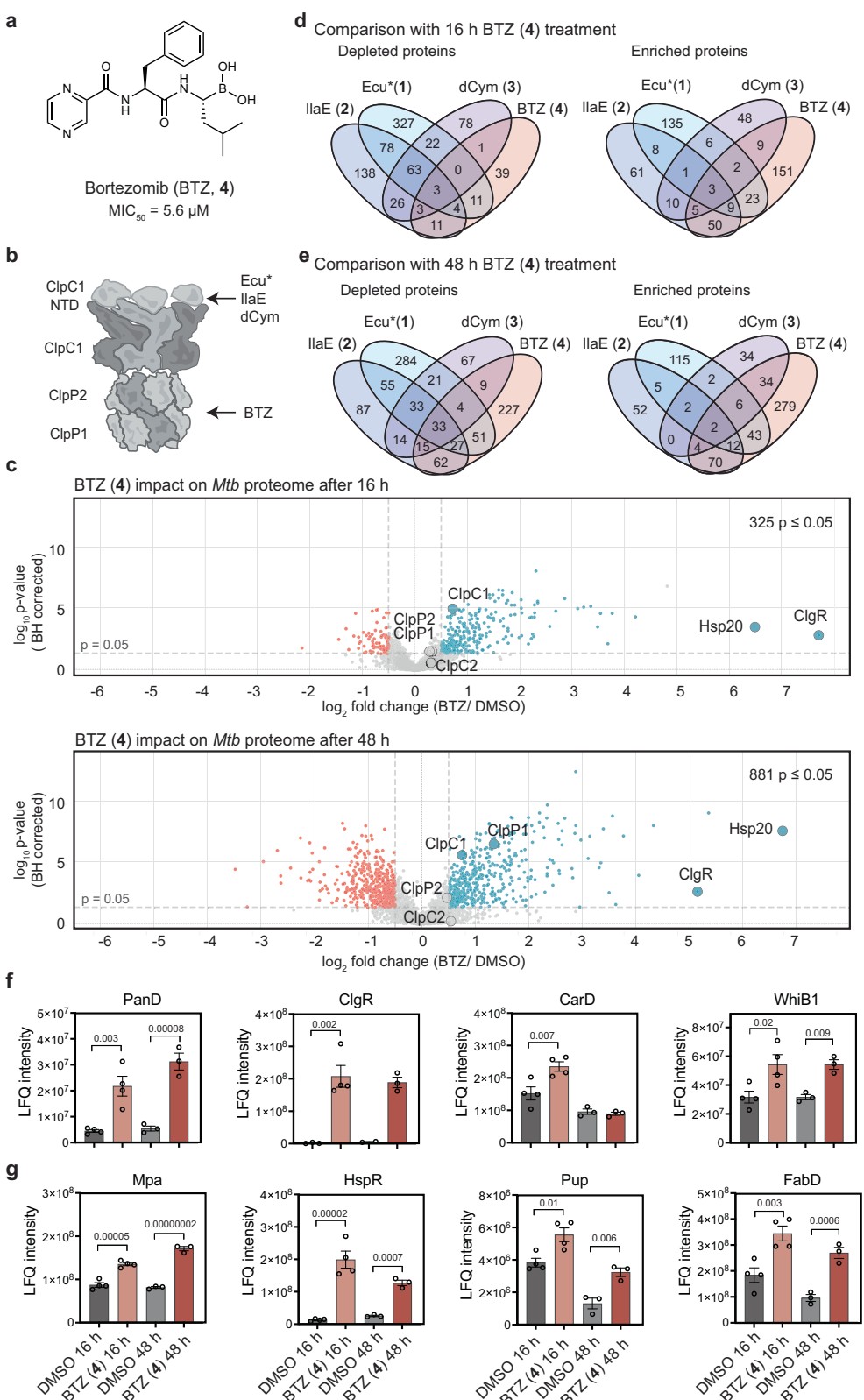

**a** Bortezomib (BTZ, **4**)
MIC$_{50}$ = 5.6 µM

**b**

**c** BTZ (**4**) impact on *Mtb* proteome after 16 h

BTZ (**4**) impact on *Mtb* proteome after 48 h

**d** Comparison with 16 h BTZ (**4**) treatment
Depleted proteins
Enriched proteins

**e** Comparison with 48 h BTZ (**4**) treatment
Depleted proteins
Enriched proteins

**f** PanD · ClgR · CarD · WhiB1

**g** Mpa · HspR · Pup · FabD

our *Mtb* proteomic datasets, Hsp20 degradation was not inhibited by these compounds in vitro (Fig. 3f, g and Supplementary Fig. 6c, d). Analytical size-exclusion chromatography showed that under assay conditions Hsp20 assembles into higher order oligomers, with species estimated to correspond to 12-, 18- and 24-mers (Fig. 3f). The unexpected degradation patterns in the presence of the NRPs may reflect

the absence of cellular factors in the in vitro experiment that influence Hsp20's oligomeric organization, as well as cofactor interactions, and other components that collectively affect its recognition by ClpC1 in cells. Overall, the NRPs appear to differentially modulate substrate recognition by the ClpC1P1P2 system. Ecu* (**1**) inhibits both β-casein and PanD degradation, IlaE (**2**) exclusively inhibits PanD degradation,

**Fig. 2 | Bortezomib inhibits protein degradation in *Mtb* H37Rv. a** Structure of Bortezomib (BTZ, **4**). Minimum inhibitory concentration (MIC) to inhibit 50% of bacteria (MIC$_{50}$) indicated below structure. Raw data from the resazurin MIC assay found in Supplementary Fig. 1d and source data provided in Source data file. **b** Schematic of ClpC1P1P2 complex with compound target site indicated, NRPs bind to the NTD of ClpC1 and BTZ (**4**) interacts with the protease. **c** Volcano plots representing the effects of BTZ (**4**) on the proteome of *Mtb* H37Rv (top = 16 h incubation, bottom = 48 h incubation). Statistical significance was calculated with a two-way ANOVA, and *p*-values were BH adjusted. Red dots represent depleted proteins (<−0.5 log$_2$ fold change, *p*-value < 0.05). Blue dots represent enriched proteins ( > 0.5 log$_2$ fold change, *p*-value < 0.05). Clp proteins and highly upregulated Hsp20 and ClgR are labelled. 16 h dataset includes *n* = 4 biological replicates and 48 h includes *n* = 3 biological replicates. **d, e** Venn diagrams showing number of similarly or uniquely affected proteins, depleted (left) or enriched (right) between NRPs (**1**–**3**) and BTZ (**4**) 16 h (**d**) and BTZ (**4**) 48 h (**e**). **f, g** Raw abundance of ClpC1 substrates (**f**) and 20S proteosome substrates (**g**) following BTZ (**4**) treatment. Data is presented as mean ± SEM, statistical significance was calculated with a BH-adjusted two-way ANOVA, 16 h *n* = 4, 48 h *n* = 3. Source Data are provided in Supplementary Data 4.

and dCym (**3**) inhibits PanD and Hsp20 degradation in vitro. These findings reinforce that, despite their overlap in binding interface to the ClpC1-NTD, the NRPs do not uniformly perturb ClpC1P1P2 proteolysis but instead may differentially alter substrate recognition and processing across the proteome.

### Induction of ClpC2 rescue mechanism is cyclomarin specific

ClpC2 expression is autoregulated via promoter binding, maintaining low basal levels under normal conditions[59]. Under proteotoxic stress, or in the presence of cyclomarin, ClpC2 preferentially engages unfolded proteins or the natural product itself, relieving promoter repression and triggering upregulation of ClpC2[33,59] (Fig. 4a). To assess whether the other NRPs could similarly interact with *Mtb* ClpC2 and compare their affinity to ClpC1-NTD, we performed surface plasmon resonance (SPR). dCym (**3**) bound both the ClpC1-NTD and ClpC2 C-terminal domain (CTD) with high affinity (ClpC1-NTD K$_D$ = 0.013 μM, ClpC2-CTD K$_D$ = 0.026 μM), as did IlaE (**2**) (ClpC1-NTD K$_D$ = 0.107 μM, ClpC2-CTD K$_D$ = 0.69 μM). In contrast, Ecu* (**1**) bound ClpC1 with high affinity (K$_{D1}$ = K$_D$ = 0.39 μM for 2:1 and 1:1 Ecu*:ClpC1-NTD binding models) and ClpC2 with weaker affinity, ClpC2-CTD K$_D$ = 43.8 μM) (Fig. 4b, c and Supplementary Fig. 7). The calculated K$_D$ values for Ecu* were lower than previously reported[34] due to the use of a recently optimised SPR protocol[60] in this work (Supplementary Fig. 7).

Interestingly, ClpC2 was only significantly enriched in the proteomic data with dCym (**3**) treatment (Fig. 4d), highlighting a compound specific modulation of ClpC2 levels. To confirm the selectivity of the interaction of dCym with ClpC2, a CRISPRi-engineered ClpC2-depleted *Mtb* strain was generated. Exposure of this strain to dCym (**3**) resulted in significantly increased bactericidal effect, consistent with ClpC2 sequestering dCym (**3**) in the control strain (Fig. 4e and g). In contrast, the antibacterial activity of Ecu* (**1**) and IlaE (**2**) were unaffected by ClpC2 depletion, indicating a minimally protective role by ClpC2 for these two NRPs (Fig. 4e and g). Together, these findings demonstrate that NRPs differ in the extent to which cellular rescue mechanisms can modulate ClpC1 dysregulation.

### Hsp20 is strongly enriched following Ecu*, IlaE and BTZ treatment

PQC is essential for mitigating proteotoxic stress, with chaperones involved in refolding proteins (DnaK/DnaJ/GrpE, GroEL/GroES), preventing aggregation (Hsp20, HspX, ClpC2), disaggregating protein complexes (ClpB, DnaK/DnaJ/GrpE), and degrading irreparable proteins (ClpC1, ClpX, Mpa) (Fig. 5a)[9]. Disruption of a key PQC component, such as the ClpC1P1P2 complex, particularly in a manner that interferes with turnover of damaged or misfolded proteins, is expected to trigger a compensatory PQC response. Analysis of chaperone abundance following NRP treatment revealed focused rather than global chaperone responses. Ecu* (**1**) treatment significantly enriched ClpC1, Hsp20, HspX and DnaJ2. IlaE (**2**) treatment led to enrichment of Hsp20 and ClpB, while dCym (**3**) treatment enriched ClpC2, GroEL1 and GroES (Supplementary Fig. 8 and Supplementary data 1). In contrast, BTZ (**4**) treatment induced a broad and coordinated PQC response, with significant enrichment of nearly all major chaperones, including ClpC1, Hsp20, HspX, DnaK, DnaJ1, DnaJ2, GrpE, ClpB, ClpX, Mpa, GroEL1,

GroEL2, and GroES at one or both timepoints (Supplementary Fig. 9 and Supplementary Data 4). This widespread BTZ-associated induction of chaperone systems is consistent with a broader blockade of proteolytic pathways, resulting in accumulation of misfolded or undegraded proteins that necessitate enhanced PQC.

Although the NRPs do not trigger the broad chaperone response observed with BTZ (**4**), Ecu* (**1**) and, to a lesser extent, IlaE (**2**) drove Hsp20 enrichment to levels comparable to BTZ (**4**). Hsp20 is a small heat shock protein associated with "holdase" activity, shielding exposed hydrophobic regions of unfolded proteins in order to buffer aggregation during proteotoxic stress[9,14]. Following the 48 h incubation, Hsp20 emerged as the most strongly induced protein across these three conditions, with inductions associated with Ecu* (**1**) and BTZ (**4**) both exceeding 90-fold (Figs. 5b, 1e, f and 2c). In contrast, dCym (**3**) had no significant effect on Hsp20, indicating that its induction is not a universal consequence of PQC-dysregulation. This pattern suggests that the robust Hsp20 response to Ecu* (**1**) and, to a lesser extent, IlaE (**2**) may represent a distinctive feature of their activity and warrants further investigation.

Given that NRPs have been reported to mimic unfolded proteins, a broad chaperone substrate class, and considering the established interaction between ClpC2 and cyclomarin, we investigated potential interactions between the NRPs and Hsp20[34,59]. SPR confirmed binding between Ecu* (**1**) and Hsp20 (K$_D$ = 3.5 μM), whereas no measurable binding was detected for IlaE (**2**) and dCym (**3**) (Fig. 5c and Supplementary Fig. 7). Despite lacking structural homology with ClpC proteins, Hsp20 exposes hydrophobic surfaces during oligomeric rearrangements that may allow interactions with NRPs[14,61]. Notably, degradation assays showed that Ecu* (**1**) did not inhibit Hsp20 turnover, and analytical size exclusion chromatography revealed no effect on Hsp20 oligomerisation in the presence of Ecu* (**1**) (Fig. 5d), suggesting that binding alone is insufficient to alter Hsp20 turnover or oligomeric state in vitro.

To distinguish transcriptional from post-translational effects, we performed unbiased RNA-sequencing of *Mtb* following Ecu*(**1**) treatment. Ecu* (**1**) dysregulated 236 genes, with 86% of these upregulated, and there was limited overlap between transcriptional effects and protein levels for individual genes, consistent with substantial post-translational regulation (Fig. 5e, f and Supplementary Data 5). The transcriptional activator ClgR is known to positively regulate the transcription of key proteins, including Hsp20, ClpC1, ClpP1, ClpP2 and ClgR itself[47,62,63]. Notably, a number of these are also substrates of the ClpC1P1P2 proteolytic complex, forming a finely tuned feedback loop that coordinates stress responses[14,47]. We investigated the specific transcript level for each of these proteins and observed uniform upregulation of the ClgR regulon (Fig. 5g and Supplementary Data 5). This may contribute to the observed enrichment of Hsp20, ClpC1, ClpP1 and ClpP2 following Ecu* (**1**) treatment. These findings emphasise the complex and multifaceted regulation of Hsp20, which is likely important for instigation of a rapid and tightly regulated response to stress.

### Compound sensitisation of *Mtb* to heat stress

Heat stress triggers the accumulation of unfolded proteins in bacteria leading to the formation of cytotoxic aggregates, which are cleared by

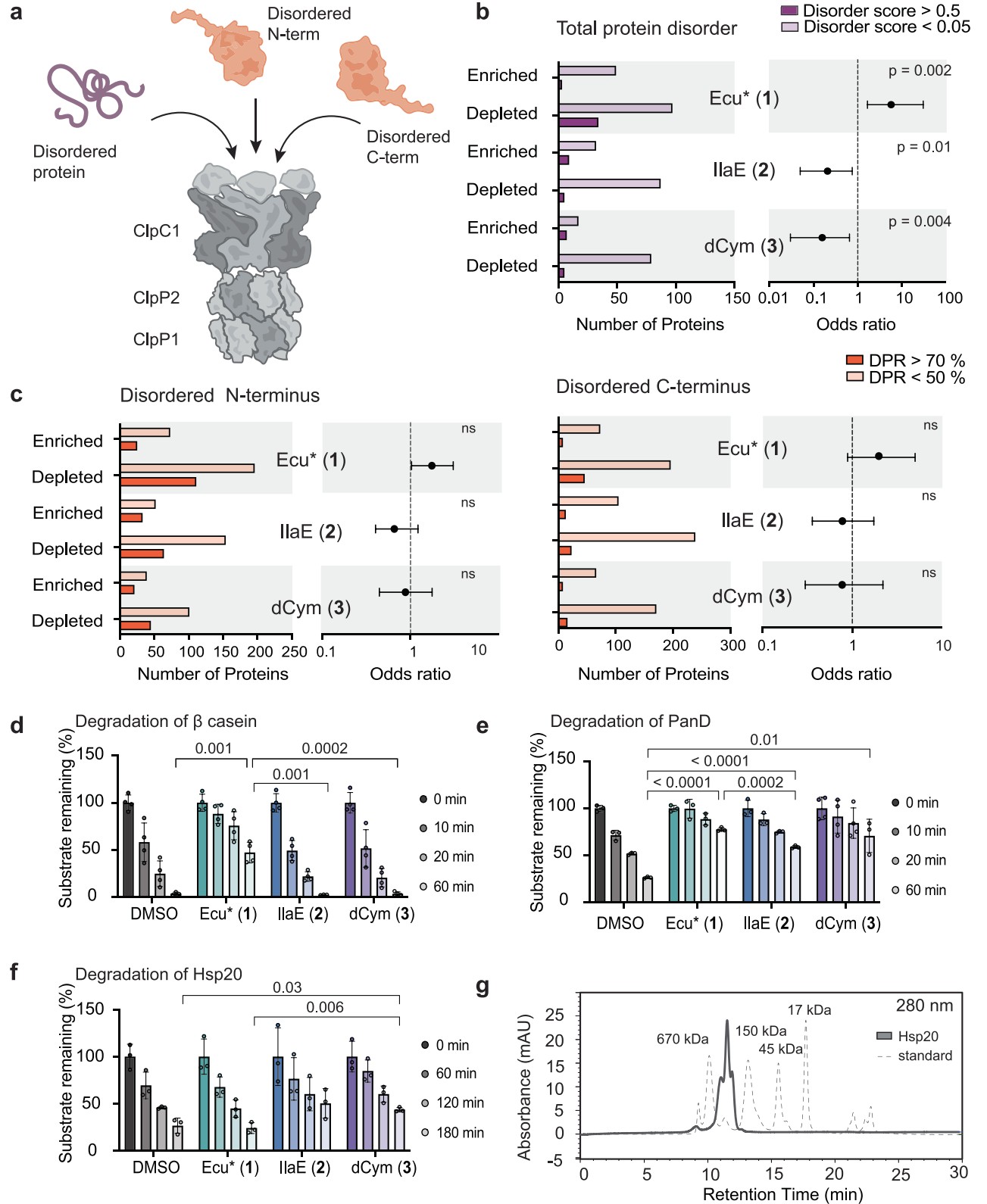

the ClpC1P1P2 proteolytic complex[64,65]. To confirm accentuated effects of heat stress on compound activity, *Mtb* was treated with the NRPs in combination with mild heat stress. Both Ecu* (**1**) and dCym (**3**) showed significantly increased bactericidal activity compared with unstressed controls (Fig. 5h). The increased activity of dCym (**3**) under heat stress is consistent with increased availability of the compound due to competition between unfolded proteins and ClpC2[34]. Given the

relatively low affinity for Hsp20 ($K_D = 3.5\,\mu M$, Fig. 5c), the enhanced antimycobacterial activity of Ecu* (**1**) under heat stress likely arises primarily from exacerbated ClpC1P1P2-mediated dysregulation of unfolded protein clearance. Whilst IlaE (**2**) treatment trended toward increased activity under heat stress, this did not reach statistical significance, suggesting its activity is less influenced by PQC dysregulation.

**Fig. 3 | NRPs have distinct impacts on substrate degradation. a** Schematic representing ClpC1P1P2 substrates with characteristic disordered termini and intrinsic disorder. **b** Impact of NRPs on intrinsically disordered proteins. The bar charts represent the number of proteins significantly enriched or depleted by each NRP, which were intrinsically disordered (AlphaFold2 predicted disorder score > 0.5, dark purple) or lacking intrinsic disorder (AlphaFold predicted disorder score <0.05, light purple). The two-sided Fisher's exact test is presented in the associated forest plots; centre of the error bar represents the odds ratio and error bars indicate the 95% confidence interval. Degree of freedom = 1, Ecu* (**1**) $n = 183$, IlaE (**2**) $n = 133$, dCym (**3**) $n = 108$ observations across all categories. Source data provided in Supplementary Table 1. **c** Influence of NRPs on abundance of proteins with disordered N-termini (top) and C-termini (bottom). The bar charts represent the number of proteins which were significantly enriched or depleted in the presence of the NRPs,

which had disordered termini ( > 70% disorder promoting residues (DPR), dark red) or non-disordered termini ( < 50% DPR, light red). The two-sided Fisher's exact test is presented in the associated forest plots; centre indicates the odds ratio and error bar represents 95% confidence interval. Degree of freedom = 1, for N-terminus disorder; Ecu* (**1**) $n = 464$, IlaE (**2**) $n = 380$, dCym (**3**) $n = 261$, and for C-terminus; Ecu* (**1**) $n = 405$, IlaE (**2**) $n = 303$, dCym (**3**) $n = 206$ observations across all categories. ns not significant. Source data are provided in Supplementary table 2. **d–f** Quantification of in vitro degradation assays. Data is presented as mean ± SD, β-casein $n = 3$, PanD $n = 3$ (DMSO, **1**, **2**), $n = 4$ (**3**), Hsp20 $n = 3$. Statistical significance between endpoints was calculated using two-tailed unpaired $t$-tests. **g** Analytical size exclusion chromatography depicting oligomeric state of Hsp20 under assay conditions. Source data are provided in Source data file.

To determine whether these effects equated to broader consequences of proteostasis impairment, we assessed the activity of protease inhibitor BTZ under the same conditions. BTZ also exhibited significantly enhanced toxicity under heat stress, reducing bacterial viability to levels comparable with those observed for Ecu* (**1**) and dCym (**3**) (Fig. 5h). Together, these results confirm that disrupting ClpC1 can impair *Mtb* survival under stress as effectively as global inhibition of mycobacterial proteases, highlighting ClpC1 dysregulation as a genuine target for future antibacterial development.

## Discussion

PQC is an essential component of the bacterial stress response[8,9,66,67] and it is vital that both substrate recognition and cellular levels of proteins involved in PQC are tightly regulated[8]. The significant and distinct changes to the *Mtb* proteome orchestrated by NRP derivatives Ecu* (**1**), IlaE (**2**), and dCym (**3**) are particularly notable considering that, under normal conditions, the abundance of most proteins is maintained within 2 orders of magnitude[68]. This indicates that the *Mtb* proteome is under tight regulation, and the variation in protein levels induced by the NRPs, particularly Ecu* (**1**), is likely to be crucial for their antimycobacterial activity. Despite all three NRPs binding to the same interface of the ClpC1-NTD, with substantial contact overlap, their effects on the *Mtb* proteome, both globally and on an individual protein level, vary significantly. This points to subtly different mechanisms dictating the modulation of ClpC1 activity, substrate recognition, and potentially the nature of the disordered substrates that each NRP mimics.

A key observation from this work is that the NRPs differentially affect *Mtb* ClpC1P1P2 substrates. Previous studies have relied primarily on model substrates, such as FITC-casein or ssrA-GFP, to profile NRP-induced dysregulation of ClpC1P1P2 proteolysis[19,69]. Assessment of the consequences for native substrates in vitro and within *Mtb* cells across each of the three NRP families in the current study provides evidence for distinct and substrate-dependent effects. The absence of statistically significant trends amongst proteins with disordered termini, a previously validated ClpC1 recognition motif[14,21], following NRP treatment within cells can be explained in part by the selective in vitro NRP-effects on proteins within this substrate class. Each NRP inhibited the degradation of PanD, confirming the capacity of these NRPs to disrupt the recognition and processing of C-terminally disordered *Mtb* proteins. However, only dCym (**3**) inhibited Hsp20 degradation, a protein that also contains a disordered C-terminus but assembles into high-order complexes. Intriguingly, this differs from previous studies that show that cyclomarin A can activate substrate degradation (albeit with different substrates and compound concentrations). This suggests that both substrate and concentration may influence whether cyclomarin acts as an activator or inhibitor[70,71]. Interestingly, our data indicate that the overall substrate architecture may influence susceptibility to NRP-induced dysregulated ClpC1P1P2 proteolysis. Each of the NRPs exhibited distinct ClpC1P1P2 dysregulation profiles using in vitro assays and within *Mtb* cells, despite substantial overlap in their binding

mode to the ClpC1-NTD interface. The results highlight the importance of studying the fate of native *Mtb* ClpC1P1P2 substrates to understand how NRPs (and future ClpC1-targeting molecules) may influence ClpC1P1P2 proteolysis. Importantly, this substrate-specific modulation provides a potential strategy for designing ClpC1-targeting antibiotics that selectively target essential mycobacterial proteins.

In this study, we show that NRPs Ecu*(**1**) and IlaE (**2**) bind the stress-induced chaperone *Mtb* ClpC2, as previously reported for dCym (**3**)[34,59]. Under non-stress conditions, ClpC2 represses its own transcription by binding its promoter[59]. Stress-induced protein unfolding redirects ClpC2 to exposed hydrophobic regions of client proteins, resulting in increased expression of ClpC2. Cyclomarin has been proposed to mimic native ClpC1/ClpC2 substrates, and by engaging ClpC2, it is sequestered from its primary target, ClpC1. This interaction reduces promoter binding and triggers ClpC2 upregulation, which further protects ClpC1 from dysregulation[34,59]. Given the structural similarity between ClpC1-NTD and the ClpC2-CTD, and similarities between the NRPs, it was plausible that ecumicin and ilamycins could also be sequestered by ClpC2. We demonstrate, however, that the interactions between Ecu* (**1**) or IlaE (**2**) with the ClpC2-CTD are insufficient to trigger ClpC2 upregulation in *Mtb* cells and so engage the canonical ClpC2 response. Therefore, in contrast to cyclomarin, ecumicin and ilamycins are unlikely to be functionally sequestered by ClpC2. This was corroborated by their unchanged antibacterial activity in a ClpC2 knockdown *Mtb* strain, whereas the antimycobacterial activity of dCym (**3**) was increased when ClpC2 was knocked down. This mechanistic distinction for Ecu* (**1**) or IlaE (**2**) is advantageous from a drug development perspective because it reveals that these compounds can modulate ClpC1 without ClpC2 interference, highlighting their potential for selective targeting of ClpC1.

Hsp20 has emerged as a key component in the PQC pathway in *Mtb*, particularly under conditions of ClpC1P1P2 dysregulation[14]. During stress, small Hsps undergo structural changes, triggered by substrate engagement, environmental factors or possibly post-translational modifications, that facilitate their rapid accumulation[61,72]. Hsp20 degradation is not directly inhibited by Ecu*(**1**), IlaE (**2**), nor BTZ (**4**) in vitro, implying that the strong accumulation following compound exposure in *Mtb* cells could not be explained solely by direct inhibition of ClpC1P1P2-facilitated Hsp20 turnover. Instead, Hsp20 accumulation likely reflects a combination of stress-induced stabilisation, engagement with accumulating aberrant proteins arising from ClpC1P1P2 dysregulation, transcriptional upregulation, and potentially even direct binding in the case of Ecu*(**1**). The strong enrichment of ClgR provides a possible explanation for the pronounced induction of Hsp20 by BTZ (**4**) and IlaE (**2**), as ClgR positively regulates Hsp20 transcription[63]. ClgR accumulation may result from inhibited ClpC1P1P2 turnover and, particularly in the case of BTZ (**4**), the accumulation of aberrant proteins that trigger a stress response[47]. In contrast, whilst Ecu*(**1**) treatment led to the upregulation of the ClgR regulon, ClgR was not enriched at the protein level, indicating that Hsp20 enrichment in the presence of Ecu* (**1**) involves additional,

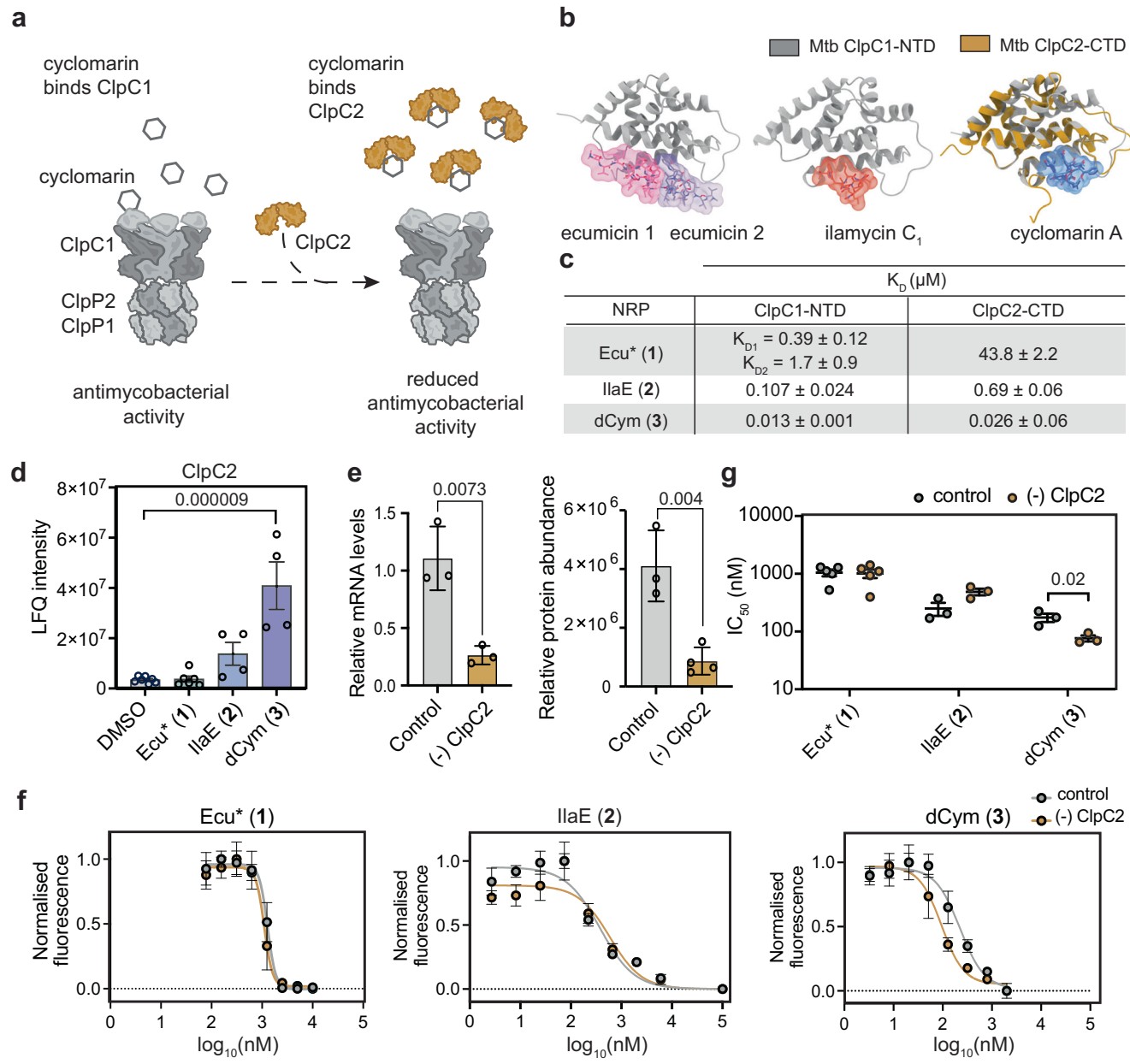

**Fig. 4 | ClpC2 is enriched in response to dCym (3). a** Schematic illustrating ClpC2 sequestration of cyclomarin. ClpC2 binding to cyclomarin reduces the cellular levels of free NRP available to deregulate ClpC1, thereby reducing antimycobacterial activity. **b** Crystal structures of NRPs bound to ClpC proteins. From left to right; *Mtb* ClpC1-NTD: Ecumicin (PDB 6PBS), *Mtb* ClpC1-NTD: Ilamycin B1 (PDB 6CN8), and superimposition of *Mtb* ClpC1 NTD: Cyclomarin A (PDB 3WDC) and ClpC2-CTD: Cyclomarin A (PDB 8AD9). **c** Surface plasmon resonance (SPR) binding affinities for interaction between NRPs and ClpC1-NTD and ClpC2-CTD. Raw data for three replicates is presented in Supplementary Fig. 7. NB: The $K_D$ for the Ecu*: ClpC1-NTD interaction was also 0.39 µM when using a 1:1 binding model. **d** LFQ intensities comparing NRP **1**–**3** effect on ClpC2. Data is presented as mean ± SEM. Each dot represents protein abundance from an individual biological replicate, DMSO $n = 7$, Ecu* (**1**) $n = 6$, IlaE (**2**) $n = 4$, dCym (**3**) $n = 4$. Statistical analysis was calculated with a one-way ANOVA with $p$-values adjusted by the BH method.

Source data provided in Supplementary Data 1. **e** Relative mRNA levels of ClpC2 knockdown (−) in *Mtb* mc²6026 determined by qPCR and protein abundance determined by LC-MS/MS-based proteomics. The control strain contained non-targeting sgRNA. Data is presented as mean ± SD and each dot represent an individual biological replicate, qPCR $n = 3$, LC-MS/MS DMSO $n = 3$, Ecu* (**1**) = 4. Statistical significance was calculated with two-tailed unpaired $t$-tests. **f** Growth inhibition curves form resazurin cell viability assay comparing NRP activity in control and ClpC2 knockdown *Mtb* mc²6026 strains. Data presented as mean ± SD from one biological replicate with three technical replicates. **g** $IC_{50}$ quantification from resazurin cell viability assays. Data is presented as mean ± SEM, dots represent data from three individual biological replicates, Ecu* (**1**) $n = 5$, IlaE (**2**) $n = 3$, dCym (**3**) $n = 3$ and analysed by two-tailed unpaired $t$-test on log-transformed data. Source data for all replicates are provided in Source data file.

ClgR-independent regulatory inputs. The direct interaction between Ecu* (**1**) and Hsp20, coupled with strong Hsp20 protein upregulation, raises the possibility of a sequestration model in which Hsp20 protects ClpC1 from Ecu* (**1**) induced dysregulation. However, the affinity measured for the Hsp20:Ecu* (**1**) interaction was substantially weaker than for ClpC1:Ecu* (**1**), making it unlikely that Hsp20 binding

interferes with the capacity of Ecu* (**1**) to interact with ClpC1 preferentially. Having established that Ecu* can interact with a chaperone lacking structural homology to ClpC1, it is plausible that it may also engage other chaperones beyond Hsp20. Although we did not observe substantial enrichments of other chaperones in our data, this possibility represents an interesting avenue for future research. The

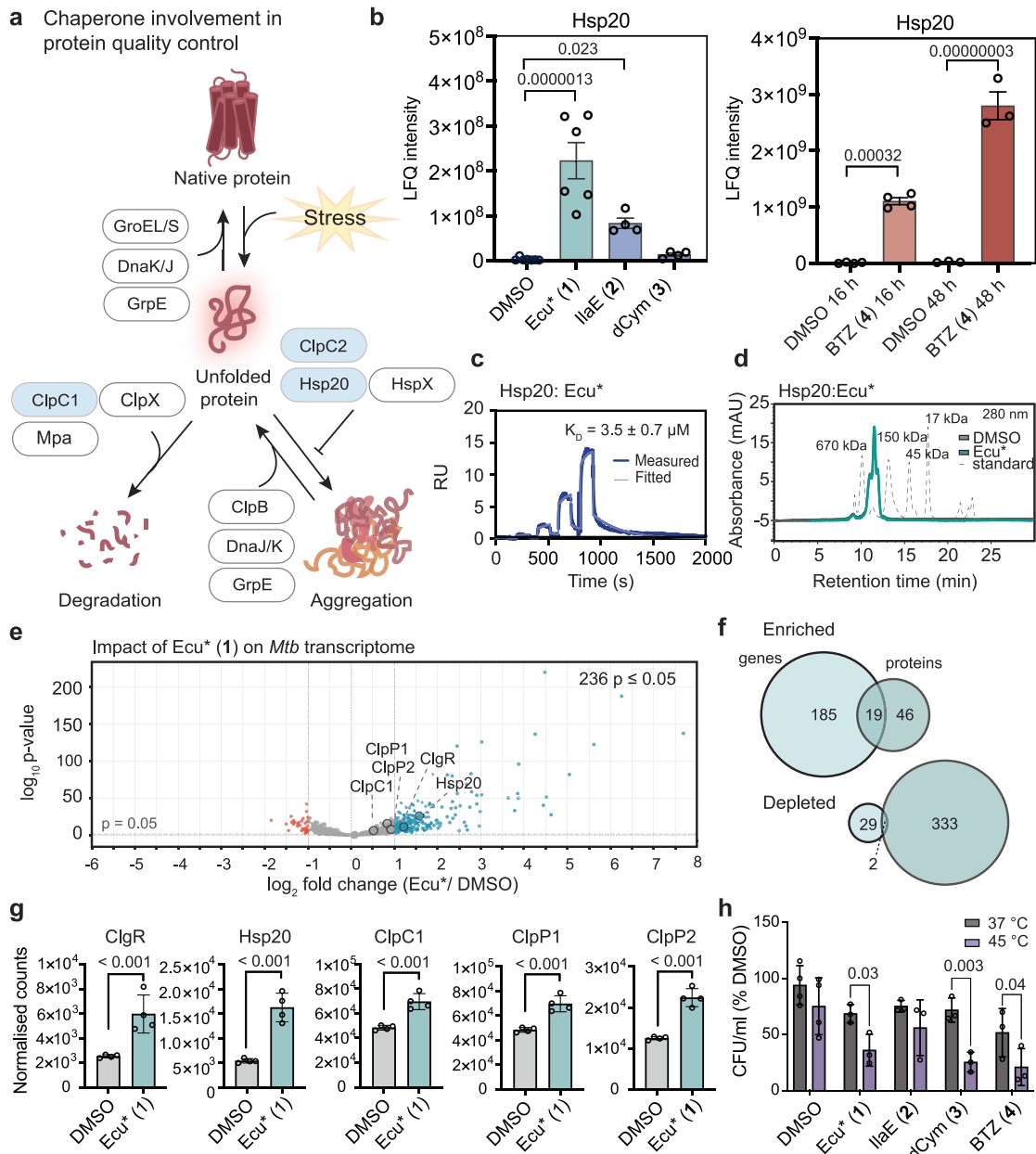

**Fig. 5 | NRPs bind stress response chaperones belonging to the PQC pathway.**
**a** Schematic representing the management of stress-induced protein unfolding by chaperones. **b** LFQ intensities comparing NRP **1**–**3** and BTZ (**4**) effect on Hsp20. Data is presented as mean ± SEM, each dot represents protein abundance from an individual biological replicate, DMSO $n = 7$, Ecu* (**1**) $n = 6$, IlaE (**2**) $n = 4$, dCym (**3**) $n = 4$, **BTZ** (**4**) 16 h $n = 4$, BTZ (**4**) 48 h. Data was analysed using a one-way ANOVA (NRPs) or two-way ANOVA (BTZ) and $p$-values were adjusted by the BH method. Source data are provided in Supplementary Data 1 and 4. **c** SPR binding experiments for interaction between Hsp20 and Ecu* (**1**). Plot represents raw data from one replicate. All three replicates are provided in Supplementary Fig. 9c. **d** Analytical size exclusion chromatography depicting oligomeric state of 25 µM Hsp20 in the presence of 50 µM Ecu* (**1**). **e** Transcriptomic analysis of Ecu* (**1**) treated *Mtb* H37Rv following 48 h incubation. Data was analysed by two-tailed Wald test, $n = 4$.

Red dots represent downregulated transcripts (≤ −1 log$_2$ fold change, $p$-value ≤ 0.05). Blue dots represent upregulated transcripts (≥ 1 log$_2$ fold change, $p$-value ≤ 0.05). **f** Venn diagrams for affected proteins across transcriptomic and proteomic datasets for Ecu* (**1**) treated *Mtb*. Enriched or depleted transcripts/proteins were those significantly dysregulated greater than ± onefold. **g** Extracted normalised counts for the transcripts of specific PQC proteins. Data presented as mean ± SD, $n = 4$ for each condition and data was analysed by two-tailed Wald test. Source data are provided in Supplementary Data file 5. **h** Colony forming unit (CFU) viability assay for heat stressed and non-heat stressed *Mtb* H37Rv treated with NRPs **1**–**3** and BTZ (**4**). Data presented as mean ± SD, $n = 3$ for each condition. Statistical significance was calculated with a two-way ANOVA. Source data are provided in Source data file.

interaction detected between Hsp20 with Ecu* (**1**) alone highlights another distinction between the NRPs and suggests that different compounds mimic distinct ClpC1-substrate characteristics, engaging and dysregulating the proteostasis machinery in unique ways.

In summary, using a combination of systematic proteomics, transcriptomics, bioinformatic analysis, and biochemical and biophysical assays, we demonstrate that, despite overlapping ClpC1 binding interfaces, each of the three NRPs exert distinct effects on the *Mtb* proteome, reflecting unique modes of ClpC1 modulation. This study indicates that overall protein architecture is a determinant of differential susceptibility to NRPs, and that the compounds affect substrate recognition in a manner that is both substrate- and NRP-

dependent. Importantly, the interactions elucidated between Ecu*(**1**) and Hsp20 and IlaE (**2**) and ClpC2 reveal that the NRPs can interact with stress-responsive chaperones, but the extent of any resulting sequestration differs markedly from that of cyclomarin with ClpC2. Indeed, we show that Ecu* (**1**) and IlaE (**2**) cause extensive proteome dysregulation at sublethal doses but do not engage the ClpC2 rescue mechanism that reduces dCym (**3**) toxicity. The ability of ecumicin and the ilamycins to bypass ClpC2 rescue, that would serve to reduce their antimycobacterial activity, makes these molecules promising molecular scaffolds for rational drug design of ClpC1-targeting ligands and as ClpC1 recruiters for next-generation BacPROTACs. Taken together, the distinct interactions, combined with the differential dysregulation of ClpC1P1P2 proteolysis, highlight the multifaceted ways NRPs perturb proteostasis. Given the essentiality of ClpC1 and PQC in *Mtb*, these findings reinforce the potential to exploit stress-adaptive pathways to enhance antimycobacterial activity and provide the foundation for the rational design of next-generation ClpC1- and PQC-targeting TB drug candidates.

## Methods

### *Mtb* H37Rv cell growth conditions

*Mtb* H37Rv cultures were grown in Middlebrook 7H9 media (Difco) supplemented with OAD (0.005% oleic acid, 0.5% bovine serum albumin, 0.2% dextrose), 0.01% tyloxapol (Sigma-Aldrich) and 0.2% Glycerol under BSL3 conditions. For all *Mtb* experiments, *Mtb* H37Rv was incubated at 37 °C, 20% oxygen and 5% carbon dioxide without agitation until desired optical density at 600 nm ($OD_{600}$) was reached. Each biological replicate was performed with a unique aliquot of a *Mtb* H37Rv glycerol stock.

### CRISPRi-mediated knockdown of ClpC2

CRISPR interference (CRISPRi) plasmids were constructed to repress ClpC2 (Rv2667) in *Mtb* mc²6026. Candidate sgRNAs (20–25 bp) were identified downstream of protospacer adjacent motifs on the non-template strand using published design principles[73]. Two sgRNAs with the highest predicted activity were chosen. The corresponding oligonucleotides (Supplementary Table 3) were assembled into the dCas9-KRAB vector backbone pJLR965 by BsmB1 golden gate cloning, following established procedures[74]. Recombinant plasmids were amplified in *E. coli*, sequence-verified by Sanger sequencing, and then introduced into *Mtb* mc26026 via electroporation. A plasmid carrying a non-targeting sgRNA (pJLR965) was included as a negative control.

### RNA extraction and qPCR analysis of knockdowns

To determine the strength of the transcriptional repression sgRNA was evaluated by growing the *Mtb* mc²6026 cultures in Middlebrook 7H9 broth supplemented with OADC (0.005% oleic acid, 0.5% Bovine serum albumin, 0.2% dextrose, 0.0085% catalase), 0.05% tyloxapol, 25 μg/mL pantothenic acid, 50 μg/mL leucine and 25 μg/mL kanamycin. Cultures were induced with 0.3 μg/mL anhydrotetracycline (ATc) and incubated at 37 °C for 10 days, with a starting $OD_{600}$ of 0.005. Total RNA was extracted via phenol-chloroform based bead-beating with 0.1 mm silica beads for three cycles of 30 s at 4800 rpm, with cooling on ice between cycles. RNA was purified using RNA Clean and Concentrator columns and treated with TURBO DNase to remove residual genomic DNA, with the absence of DNA contamination confirmed by PCR. Complementary DNA (cDNA) was synthesised using Superscript IV VILO Master Mix, incubated at 25 °C for 10 min, 50 °C for 10 min and 85 °C for 5 min. qPCR was performed in 384-well plates on a ViiA7 thermocycler using PowerUp SYBR Green Master Mix, with 3 μL of undiluted cDNA and 500 nM of each primer (Supplementary table 3). Primer concentrations and cDNA inputs were optimised to ensure linear amplification. Transcript levels were normalised to the housekeeping gene sigA and quantified using the ΔΔCt method[75]. Knockdown efficiencies were calculated relative to non-targeting sgRNA

controls, with three independent biological replicates, each measured in technical triplicate.

### Minimum inhibitory concentration (MIC) assay

A resazurin assay was used to determine the minimum inhibitory concentration of each of the NRPs. *Mtb* H37Rv was incubated until $OD_{600}$ 1.0, then diluted to $OD_{600}$ 0.002 in Middlebrook 7H9 media. *Mtb* mc²6026 was cultured in Middlebrook 7H9 broth supplemented with OADC, 0.05% tyloxapol, 25 μg/mL pantothenic acid, 50 μg/mL leucine and 25 μg/mL kanamycin. Cultures were induced with 0.3 μg/mL ATc and incubated at 37 °C until early mid-log phase and then diluted to 0.002 in Middlebrook 7H9 media, OADC, 0.05% tyloxapol, 25 μg/mL pantothenic acid and 50 μg/mL leucine. Diluted cells (100 μL) were inoculated into each well of a 96-well plate. Sequential concentrations of each compound were pipetted into each well through two-fold serial dilutions with a starting concentration of 10 μM. The plates were incubated for four days at 37 °C without agitation. Resazurin (Sigma-Aldrich) was added to a final concentration of 0.003 % (w/v) to each well and plates were incubated for 24 h at 37 °C. Fluorescence was measured (Ex/Em: 560 nm/590 nm) in a POLARstar Omega microplate reader (BMG LABTECH). Background fluorescence was subtracted from wells without cells and fluorescence was normalised to wells without compound. A sigmoid curve was fitted with non-linear regression to the dose response data. Each experiment was performed in triplicate. The minimum concentration required to inhibit 50% and 90% of bacterial growth was calculated using GraphPad Prism (version 10).

### Cell viability assay

Colony forming unit (CFU) assays were performed to measure cell viability following NRP treatment. Early, exponentially growing *Mtb* samples ($OD_{600}$ ~ 0.45) were treated with the $MC_{90}$ of the natural products and incubated for either 16 h or 48 h at 37 °C or 45 °C for heat stressed samples. The cultures were then diluted tenfold serially in a 96-well plate with Middlebrook 7H9 media. 100 μL of diluted bacteria was placed on each quadrant of a Middlebrook 7H10 agar plate (Difco) such that a range of $10^3$–$10^1$ CFU/100 μL were expected. Individual colonies were counted following a 21-day incubation at 37 °C. The ratio between the number of individual colonies persisting following treatment and the number of individual colonies present in the vehicle control was calculated as the %CFU.

### Protein expression and purification

Mycobacterial proteins *Mtb* ClpC1-NTD$_{1-148}$, *Mtb* ClpC2-CTD$_{94-252}$, *Mtb* Hsp20, *Mtb* full length ClpC1, *Mtb* ClpP1, *Mtb* ClpP2 and *Mtb* PanD were expressed in *Escherichia coli*. ClpC1-NTD-His$_6$, full length ClpC1, ClpP1-His$_4$, ClpP2-His$_4$ were cloned as described previously[34]. DNA gBlocks for ClpC2-CTD-His$_6$, His$_6$-Hsp20, and His$_6$-PanD were cloned into pET28a plasmids (Supplementary table 3), then transformed into *E. coli* DH5α competent cells. Plasmids for ClpC1-NTD-His$_6$, ClpC2-CTD-His$_6$, His$_6$-Hsp20, ClpP1-His$_4$, ClpP2-His$_4$, and His$_6$-PanD were transformed into *E. coli* BL21 (DE3), and untagged ClpC1 was transformed into Rosetta DE3 cells. Large scale expression was carried out in LB media supplemented with respective antibiotic. Cultures were incubated at 37 °C until $OD_{600}$ ~ 0.8 and expression was induced with 0.5 mM isopropyl β-thiogalactopyranoside. Expression cultures were transferred to 18 °C and incubated for a further ~20 h. Cells were harvested by centrifugation at 5000 × *g* at 4 °C for 30 min. Pellets were resuspended with lysis buffer (10 mM Tris pH 7.5, 150 mM NaCl, 1× cOmplete protease inhibitor EDTA-free (Roche) and lysozyme) and lysed with sonication.

Recombinant His-tagged proteins were purified from lysate by nickel-nitrilotriacetic acid (Ni-NTA) affinity chromatography and fractions were analysed by sodium dodecyl sulfate-polyacrylamide gel electrophoresis (SDS-PAGE). Discrete fractions of the correct molecular weight were pooled, concentrated, and loaded onto a Äkta Pure

purification system (Cytiva) with a size exclusion chromatography (SEC) column (Hiload, Superdex200 10/300, GE Healthcare) equilibrated with SEC buffer containing 50 mM HEPES-NaOH pH 7.5, 300 mM NaCl, 10% glycerol. Purification was monitored by SDS-PAGE. Discrete fractions were pooled, concentrated, flash frozen with liquid nitrogen, and stored at −80 °C. Purification of untagged ClpC1, ClpP1-His$_4$ and ClpP2-His$_4$ was performed as previously described[34]. For the generation of mature ClpP1P2 complex for degradation assays, 50 μM ClpP1-His$_4$ and 50 μM ClpP2-His$_4$ were incubated at room temperature overnight in the presence of 1 mM activator peptide Z-Leu-Leu-H (benzyloxycarbonyl-ʟ-Leucyl-ʟ-Leucinal) (PeptaNova). The mature ClpP1P2 was separated from the activator using a PD-10 desalting column (GE Healthcare) and verified by SDS-PAGE[76].

### Analytical size exclusion chromatography (aSEC)
To monitor the oligomerisation of *Mtb* Hsp20 alone and in the presence of Ecu*, 25 μM Hsp20 was premixed with 50 μM Ecu* (**1**) or equivalent volume DMSO. Samples were centrifuged and 10 μL supernatant was loaded using a Thermo Fisher U3000 BioRS UHPLC system onto an AdvancedBioSEC column 7.8 × 300 mm 2.7 μm particles 300 Å pores column (Agilent), pre-equilibrated and run with buffer containing 50 mM HEPES pH 7.4, 100 mM KCl, 10% glycerol. All runs were performed at 25 °C with a 0.5 mL/min flow rate. Data was visualised in Chromeleon (version 6.8).

### In vitro *Mtb* ClpC1P1P2 degradation assay
The degradation of model and putative ClpC1P1P2 substrates in the presence of the NRPs was monitored using in vitro degradation assays as described previously[76]. Reactions contained hexameric *Mtb* ClpC1 (0.5 μM), mature ClpP1P2 tetradecamer (0.25 μM) and substrate (15 μM β-casein, 17 μM PanD or 5 μM Hsp20) in assay buffer (50 mM HEPES pH 7.4, 100 mM KCl, 10 mM MgCl2, 10% (v/v) glycerol). An ATP regeneration system was also included, consisting of 20 mM phosphoenolpyruvate and 10 U/mL pyruvate kinase (Sigma-Aldrich). Prior to initiation, 100 μM of each compound (**1–4**) was added to the reaction mixture with a final concentration of 1% DMSO across all assays. Reactions were initiated with 5 mM ATP and incubated at 37 °C. Timepoint aliquots (5 μL) were collected and quenched with SDS sample buffer and subsequently analysed by SDS-PAGE followed by Coomassie staining. Gels were imaged with a ChemiDoc MP (BioRad Laboratories). Band intensities were quantified in ImageJ (version 1.54p) and end-point statistical significance was calculated using unpaired *t*-tests in GraphPad Prism (version 10).

### Surface plasmon resonance (SPR) assay
SPR was used to measure the binding affinities of the NRPs **1–3** to recombinant *Mtb* proteins. SPR measurements were carried out using a Biacore™ T200 (GE Healthcare), and data analysis undertaken using the Biacore Insight Evaluation Software. Experiments were performed in single cycle kinetics mode at 25 °C. The sensor chip surface of a CM5 chip (Cytiva) was activated using a 1:1 mix of 1-ethyl-3-(3-dimethylaminopropyl) carbodiimide hydrochloride and *N*-hydroxysuccinimide. ClpC1 NTD, ClpC2 NTD and Hsp20 proteins (20 μg/mL) in sodium acetate buffer (10 mM, pH 4·5) were immobilised to a target density of ~1500 RU. Ethanolamine (1 M) was used to quench any remaining *N*-hydroxysuccinimide esters on the surface of flow cells. A solution composed of 25 mM Tris, 150 mM NaCl, 0.1% DMSO (v/v), 0.01% Tween-20 (v/v) at pH 7.5 was used as the running buffer. Peptide macrocycles were injected as a concentration series with a contact time of 120 s and a dissociation time of 1200 s at a flow rate of 50 μL/min. Data were then reference subtracted and blank subtracted prior to processing. Equilibrium dissociation constants ($K_D$) were calculated by nonlinear least-squares fitting to a simple 1:1 Langmuir binding isotherm, as exploited in the Biacore T200 Evaluation software. Data are expressed as the mean ± SEM of three technical replicates. For Ecu*

(**1**) binding to ClpC1, equilibrium constants ($K_D$) were calculated assuming 2:1 binding[30] in the Biacore T200 evaluation software (kinetic heterogeneous ligand model).

### RNA isolation, library preparation, and sequencing for RNA-seq
Exponentially growing *Mtb* H37Rv cultures were treated with MIC$_{90}$ Ecu* (**1**) and incubated for a further 48 h at 37 °C. The *Mtb* cells were then centrifuged at 3725 × *g* for 10 min to pellet bacteria. Pellets were resuspended with 1 mL TRIzol and transferred to a 2 mL O-ring tube with 250 μL < 106 μm acid-washed glass beads. To disrupt the mycobacterial cell envelope, cells were subjected to six cycles of bead beating (20 s at 4600 rpm and 40 s on ice) with a (Biopec products) mini beadbeater. Samples were then clarified at 18,000 × *g* for 1 min. After a short room temp incubation, 200 μL of chloroform was added to each sample and samples were inverted by hand for 15 s and incubated at room temperature for 3 min. Samples were then centrifuged at 12,000 × *g* for 15 min at 4 °C. The upper aqueous phase containing RNA was collected. RNA was purified from the aqueous phase with the Isolate II RNA micro kit. Libraries were prepared using the (Illumina) Total RNA library kit with Ribo-Zero Plus chemistry for rRNA depletion and sequenced on an (Illumina) NovaSeq 6000 platform to generate 150 bp paired-end reads (> 65 million pairs per sample).

### RNA-seq preprocessing and analysis
Raw paired-end FASTQ files were quality filtered and adapter-trimmed using *fastp*[77] with default settings. Trimmed reads were quantified directly against an *Mtb* H37Rv reference transcriptome supplemented with rRNA/tRNA features (gentrome index) using *Salmon* (version 1.10.3)[78] in selective-alignment mode with sequence- and GC-bias correction enabled. Transcript-to-gene mapping used the H37Rv transcript FASTA, and non-coding features (rRNA, tRNA, tmRNA, sRNA) were excluded from downstream gene-level analyses. Gene-level counts were imported into R using *tximport*[79] with counts scaled by transcript length and TPM. Differential expression analysis was performed with *DESeq2*[80]. Lowly expressed genes (fewer than 10 counts in at least half of samples) were filtered. Differential expression contrasts were fitted with the Wald test, and log2 fold changes were shrinkage-estimated using *apeglm*[81]. Genes were considered significantly differentially expressed at adjusted $p < 0.05$ and absolute log 2 fold change ≥ 1.

### *Mtb* treatment and cell lysis for proteomic analysis
*Mtb* H37Rv and pre-depleted *Mtb* mc$^2$6026 was grown until mid-log phase (OD$_{600}$ ~ 0.5). For proteomics experiments, *Mtb* was split into separate flasks containing 5 mL of bacterial culture and subsequently treated with a sub-lethal dose of each compound (NRP (MIC$_{90}$), BTZ (MIC$_{50}$)) or vehicle control, DMSO. Flasks were incubated for 48 h at 37 °C without agitation. For heat stress experiments, mid-log phase *Mtb* was treated with MIC$_{90}$ of NRPs or DMSO and heat stress was induced with a 16 h incubation at 42 °C whilst controls were incubated at 37 °C for 16 h. For proteomics experiments cells were pelleted by centrifugation at 3750 × *g* for 10 min. The supernatant was removed and cell pellets were washed in PBS followed by resuspension in 200 μL of SDS lysis buffer (4% SDS, 10 mM EDTA, 1× cOmplete protease inhibitor EDTA-free (Roche) in phosphate-buffered saline). Lysed cells were heated to 95 °C in a water bath for 30 min to ensure complete sterilisation and were then frozen overnight (−20 °C). Samples were thawed at 95 °C and sonicated for 10 min (30 s on, 30 s off) at 70% amplitude at room temperature using a QSonica Q800R2. Lysates were clarified at 18,000 × *g* for 10 min at room temp and protein concentration was determined by BCA total protein assay (Pierce).

### Sample preparation for LC-MS/MS analysis
Lysates were thawed at 95 °C and SDS was removed by chloroform/methanol protein precipitation[82]. For 100 μL of lysate; 400 μL

methanol, 100 μL chloroform and 300 μL water was added to each sample and vortexed after each addition. Samples were clarified at $9000 \times g$ for 5 min at room temperature and the top 80% of the upper phase was aspirated. An additional 300 μL of methanol was added, followed by clarification at $9000 \times g$ for 5 min. The entire supernatant was aspirated and the pellet allowed to dry briefly before resuspension in SDC buffer (4% sodium deoxycholate, 0.1 M Tris-HCl pH 8.0) for 10 min at 95 °C with mixing at 1000 rpm in a Thermomixer-C (Eppendorf). Protein concentration was quantified by BCA total protein assay (Pierce). 20 μg of protein was reduced and alkylated by adding 10 mM TCEP, 40 mM chloroacetamide for 10 min at 95 °C with mixing at 1000 rpm in a Thermomixer-C (Eppendorf). The SDC concentration was diluted to 1% with water and the samples were cooled to room temperature. *Mtb* lysates were digested with 200 ng trypsin (Promega) and incubated for 16 h at 37 °C with 1000 rpm shaking in a Thermomixer-C (Eppendorf).

SDB-RPS StageTip peptide clean-up was performed prior to LC-MS/MS as described previously[83]. An equal volume of 1% TFA in ethyl acetate was added to the digests and vortexed. SDB-RPS StageTips were constructed with 200 μL pipette tips packed with doubly stacked SDB-RPS discs (Sigma-Aldrich Cat No. 66886-U), punched with an 18-gauge needle. The StageTips were mounted into a custom 3D printed holder over a 96-well polypropylene waste plate[83]. Each tip was wetted with 100 μL of acetonitrile, centrifuged at $1000 \times g$ for 1 min and the flowthrough discarded. After wetting, tips were washed with 100 μL of 30% methanol, 1% TFA in $H_2O$ and centrifuged at $1000 \times g$ for 3 min. Tips were equilibrated with 100 μL of 0.2% TFA in $H_2O$ and centrifuged for 3 min at $1000 \times g$. The bottom phase of the digests were loaded onto the tips and centrifuged for 5 min at $1000 \times g$. Loading was followed by two wash steps, both with 1% TFA in ethyl acetate with centrifugation for 3 min at $1000 \times g$. The last wash consisted of 100 μL of 0.2% TFA in $H_2O$ and centrifuged for 3 min at $1000 \times g$. The flowthrough was discarded after each step. The waste plate was exchanged for a bottom plate containing a holder supporting an unskirted PCR plate. Peptides were eluted with 5% ammonium hydroxide and 15% $H_2O$ in acetonitrile and centrifuged for 5 min at $1000 \times g$. The eluent was dried until <1 μL remained using nitrogen drying with a TurboVap 96 Dual (Biotage), with both plate and gas temperatures set to 65 °C and the nitrogen gas flow-rate set to 40 L/min. The peptides were immediately resuspended in 20 μL of 5% (v/v) formic acid and the protein concentration was measured with a Direct Detect spectrometer (Millipore). Samples were stored at 4 °C until ready to be analysed by LC-MS/MS.

## LC-MS/MS analysis

Using a Vanquish Neo UHPLC system, peptides (0.5 μg) in 5% (v/v) formic acid were injected (3 μL) directly onto a 45 cm × 75 μm C18 (Dr. Maisch, Ammerbuch, Germany, 1.9 μm) fused silica analytical column with a ~10 μm pulled tip, coupled to a nanospray ion source. Peptides were resolved over a gradient of 5% acetonitrile to 40% acetonitrile over 90 min with a flow rate of 0.3 μL/min. Peptides were ionized by electrospray ionization at 2.4 kV. Tandem mass spectrometry analysis was carried out on an Orbitrap Exploris 480 (Thermo Fisher Scientific), an Orbitrap Fusion Lumos Tribrid (Thermo Fisher Scientific) or a Q Exactive Plus Hybrid Quadrupole-Orbitrap (Thermo Fisher Scientific). MS1 spectra were generated in a 350–1650 $m/z$ mass range at a 30,000 orbitrap resolution with a maximum injection time of 54 ms. Precursors were selected for MS2 analysis in a data independent acquisition (DIA) using 20 variable width isolation windows or data dependent acquisition (DDA), where the top 10 ions were selected for fragmentation in each MS1 scan. Higher-energy induced dissociation (HCD) fragmentation was used with stepped collision energies of 25, 27 and 30 for DIA analysis or fixed HCD of 27 for DDA analysis. The orbitrap was operated at 30,000 resolution, with and automatic gain control of $1 \times 10^6$ and a maximum injection time of 54 ms.

## LC-MS/MS data and statistical analysis

Raw DIA data was analysed in DIA-NN[84] (version 1.8) and DDA data was analysed in MaxQuant (version 1.6.3.4). The DIA NN and MaxQuant outputs have been uploaded to the ProteomeXchange consortium member EBI-PRIDE (identifier: PXD057335). The precursor false discovery rate was set to 1%, and the *Mtb* H37Rv UniProt database downloaded on 14th April 2022 was supplied for peptide identification that contained 4062 protein sequence entries. Library generation was enabled with charge states 1–4, precursor $m/z$ from 350 to 1650. Carbamidomethylation of Cys, was searched as a fixed modification and oxidation of Met, N-terminal excision of Met and acetylation of protein N-termini were searched as variable modifications. Likely interferences were removed and match between runs was enabled for DDA data. Enzymatic specificity was set to trypsin with a maximum of 2 missed cleavage permitted and minimum peptide length of seven amino acids. Statistical tests on DIA NN output were performed using the R software package (version 4.2.1). To detect outliers, the mvoutlier (Multivariate Outlier detection Based on Robust methods) package was used and the sign2 function was applied. One outlier was detected and was excluded from all further statistical analysis. Fold changes for protein abundance were calculated using the median of each treatment. Significance between the vehicle control and NRP treated *Mtb* was calculated using one-way ANOVA. The resulting $p$ values were adjusted to control for multiple testing using the Benjamini–Hochberg correction. Significance was set at $p < 0.05$, corresponding to a false discovery rate in the ANOVA of 5%. Processed data were plotted using Tableau (version 24.1).

## Gene ontology and enrichment analysis

Gene ontology enrichment analysis was performed to identify biological processes modulated by the NRPs. Proteins that presented a significant ($p < 0.05$) fold change > 0.5 following NRP treatment were considered differentially abundant. Differentially abundant proteins were grouped as either increasing or decreasing, and the two datasets for each NRP were imported into STRINGdb (version 12.0)[85] for gene ontology enrichment analysis. The level of protein annotation in *Mtb* is relatively low with 15% *Mtb* H37Rv proteins lacking any gene ontology annotation. To enable identification of enrichments by STRINGdb, these proteins were removed from the dataset. *Mtb* H37Rv was set as the reference organism. Enrichment strengths were calculated as the ratio between the number of proteins annotated with a term in the submitted dataset and the number of proteins expected to be annotated with the term in a random dataset of the same size. The false discovery rate cut-off was 5% and p-values were corrected for multiple testing with the Benjamini-Hochberg procedure. Clustering of gene ontology enrichments with associated false discovery rates was performed using Revigo (version 1.8.1)[86] with default settings and *Mtb* H37Rv selected as the species.

## Disorder prediction and statistical analysis

To assess the correlation between termini disorder and differential abundance following NRP treatment, each protein in the *Mtb* proteome was assigned a predicted termini disorder score. Disorder promoting residues (DPR) included Ala, Arg, Gly, Gln, Ser, Pro, Glu and Lys. The ratio of the number of DPR in the 15 N or C-terminal amino acids and number of terminal amino acids (15) were calculated as percentages. Proteins with a percent DPR > 70% were considered to have disordered termini and <50% were not considered to have a disordered terminus[14] (Supplementary Data 3).

Total intrinsic disorder scores were also assigned to each protein in the *Mtb* proteome using the AlphaFold-disorder prediction embedded in mobiDB (version 5.0) platform[57]. The AlphaFold-disorder prediction processes PDB files generated by AlphaFold2, considering the pLDDT score and the Relative Solvent Accessibility[58].

To assess the relationship between NRP-induced differential abundance and termini disorder or total disorder, the Pearson's $\chi^2$ test was performed using the R software package (version 4.2.1). The statistic compares the frequencies observed in certain categories to the frequencies expected in those categories by chance. The $\chi^2$ test was accompanied by Yates' continuity correction to moderate underestimation of Pearsons's $\chi^2$ $p$ values. The Fisher's exact test was performed to calculate the exact probability of the $\chi^2$ statistic to support potentially inaccurate $\chi^2$ approximations due to small sample sizes. The Fisher's exact test was considered true if the 95% confidence interval of the odds ratio did not intercept 1.

### Chemical synthesis of peptide natural products and derivatives

Ecu* (**1**) was synthesised according to protocols previously described by Hawkins et al.[36] dCym (**3**) was synthesised according to the procedure described in Barbie and Kazmaier[24] and in Junk et al.[33] The synthesis of this compound has also been previously reported by Kiefer et al.[87] IlaE (**2**) was also prepared by chemical synthesis, using a slightly-modified protocol to that reported by Bedding et al.[38] Synthetic procedures and characterisation data are provided in the Supplementary Information.

### Statistics and reproducibility

A sample size of $n \geq 3$ biological replicates was chosen for robust statistics. No statistical method was employed to predetermine sample size. No data were excluded besides one Ecu* (**1**) treated *Mtb* LC-MS/MS dataset, which was excluded from statistical analysis due to a suspected experimental anomaly. The experiment was performed a total of 7 times and 6 were included in statistical analysis. The experiments were not randomised, and the investigators were not blinded to allocation during experiments and outcome assessment.

### Reporting summary

Further information on research design is available in the Nature Portfolio Reporting Summary linked to this article.

## Data availability

All data supporting the results of this study are presented within the article, the Supplementary Information file and Supplementary Data files. Source data are provided with this paper. Additional information can be obtained from the corresponding authors upon request. The mass spectrometry proteomics data have been deposited in the ProteomeXchange Consortium with the dataset identifier PXD057335. The raw sequence reads have been deposited in the NCBI SRA under PRJNA1346855. Processed data files (normalized counts and VST) have been deposited in the GEO NCIB-NIH repository with the dataset identifier GSE310713. Source data are provided with this paper.

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

## Acknowledgements

We acknowledge the Australian Research Council Centre of Excellence for Innovations in Peptide and Protein Science (CE200100012 to R.J.P.) for funding. We thank the National Health and Medical Research Council for funding (APP1153493 and APP2011467 to W.J.B.). R.J.P. was also funded by a National Health and Medical Research Council Investigator Leadership Fellowship (APP1174941). J.L. is funded by ESP 467 (Grant-DOI 10.55776/ESP467) of the Austrian Science Fund (FWF). The authors also acknowledge the Research training program (I.K.B., M.J.B., J.W.C.M. and P.H.) and John A. Lamberton Research Scholarship (I.K.B. and J.W.C.M.) for PhD funding. We are grateful to Uli Kazmaier, Lukas Junk (Saarland University) and Boehringer Ingelheim for gifting the dCym compound. We thank Nandan Deshpande (University of Sydney) from Sydney Informatics Hub for Bioinformatics support. We also thank SydneyMS, The University of Sydney for Mass spectrometry instrumentation and Angela Connolly and Denise Tran from Sydney MS, The University of Sydney for technical support.

## Author contributions

R.J.P., M.L., W.J.B., T.C. and I.K.B. designed experiments. M.J.B and P.H. performed chemical synthesis and analysis of ilamycin E and the ecumicin analogue. I.K.B., M.S., T.W. and D.Q. performed microbiological assays, including those under BSL3. I. K. B and J. L performed degradation assays. M.B.N., W.J.J., and G.M.C. generated *Mtb* knockdown strains. M.F., D.H. and I.K.B. performed RNA sequencing. I.K.B. and M.L performed mass spectrometry-based proteomics and analysis. J.C.W.M. and M.J.B. performed SPR experiments. I.K.B., M.J.B., M.L., R.J.P. and W.J.B. prepared the manuscript with input from all authors.

## Competing interests

The authors declare no competing interests.

## Additional information

[1]School of Chemistry, The University of Sydney, Sydney, NSW, Australia. [2]Australian Research Council Centre of Excellence for Innovations in Peptide and Protein Science, The University of Sydney, Sydney, NSW, Australia. [3]Research Institute of Molecular Pathology (IMP), Vienna, Austria. [4]Tuberculosis Research Program at the Centenary Institute, The University of Sydney, Sydney, NSW, Australia. [5]Department of Microbiology and Immunology, University of Otago, Dunedin, New Zealand. [6]Centenary Institute and Faculty of Medicine and Health, The University of Sydney, Sydney, NSW, Australia. [7]Central Clinical School, Faculty of Medicine and Health, The University of Sydney, Sydney, NSW, Australia. [8]Charles Perkins Centre and School of Medical Sciences, Faculty of Medicine and Health, The University of Sydney, Sydney, NSW, Australia. ✉e-mail: warwick.britton@sydney.edu.au; mark.larance@sydney.edu.au; richard.payne@sydney.edu.au

