## [Transparent Peer Review file · Nature Communications]

ClpC1-targeting peptide natural products differentially dysregulate the proteome of *Mycobacterium tuberculosis*

Corresponding Author: Professor Richard Payne

Version 0:

Reviewer comments:

Reviewer #1

(Remarks to the Author)

This is an interesting article, that explores a very important question concerning the mode of action of MtbClpC1 antibiotics. Very little is known on how ecumicin, cyclomarin and rufomycin lead to cell death, particularly the death of persisters. While the approach is interesting, demonstrating the power of modern proteomics, the results obtained are mostly descriptive and fail to explain the mechanistic differences between these 3 antibiotics. Following this article we will know which substrates are up or down regulated by each antibiotic but very little on how this happens. The statement that the differences between the different antibiotics are the result of specific binding of ecumicin or rufomycin to particular chaperones is not fully supported by the data presented, at least in the current version of the article. Indeed, to really understand the differences between antibiotics it would require extensive mechanistic studies that are totally absent from the work.

Minor Point

There are multiple previous articles using the terminology rufomycin instead of ilamycin. It is true that the authors explain that ilamycins are the same as rufomycins, but perhaps this should be done earlier in the text.

Major Points

Major Point 1:

The authors state:

"Ecu*(1) bound to MtbClpC1NTD with a KD of 8.58 μ M, IlaE(2) a KD of 0.11 μ M and dCym(3) a KD of 0.013 μ M. These values are consistent with those reported in previous studies 41."

We believe that the reported values for ecumicin and rufomycin are significantly lower as for example in the following article (that is not cited here), "High-Resolution Structure of ClpC1-Rufomycin and Ligand Binding Studies Provide a Framework to Design and Optimize Anti-Tuberculosis Leads" by Wolf et al, the values described for ecumicin and rufomycin (also using SPR) are very different. For ecumicin binding to ClpC1-NTD the KD value was described as 50 nM and for rufomycin the value was 63 nM. The KD for ecumicin in that previous publication is therefore circa 200x lower than the one you describe here and that is even more puzzling when you state here the following: "a synthetic ecumicin analogue (Ecu*, 1) with a simplified amino acid composition, and superior potency, to the parent natural product". So either your ecumicin analog is in fact less potent than you propose or the KD value you present here is not correct. This paradox is even more puzzling when you report a MIC90 of 0.15 μ M for ecumicin (supplementary Fig1) which was used in the proteomics experiments.

Of note, if the ecumicin analog is not as potent as you propose, it would be perhaps important to do this kind of studies with the more potent compound.

Major Point 2:

“Increased temperature has been shown to increase the bactericidal effect of cyclomarin; this is thought to be due to occupation of ClpC2 by unfolded proteins, thus allowing cyclomarin to act on ClpC1 41.”

Another reason for this high temperature effect, which is more likely in our view, is that high temperature increases the burden of unfolded proteins, challenging the protein quality system and therefore making the cell more sensitive to its dysregulation by antibiotics. To test the author’s hypothesis, it would be important to inhibit just ClpP1P2 (for example with bortezomib), this way bypassing ClpC1, and check if you would obtain similar results with temperature increase. If not, your hypothesis would be supported.

Major Point 3:

The authors cite and refer multiple times to the article “The unfoldase ClpC1 of *Mycobacterium tuberculosis* regulates the expression of a distinct subset of proteins having intrinsically disordered termini” by Lunge et al. In this specific publication the authors show that ClpC1 is responsible for the degradation of Hsp20, it is clearly stated:

“Further, we observed that Hsp20 is poorly expressed in WT Mtb and that its expression is greatly induced upon depletion of clpC1 or clpP2. Remarkably, high Hsp20 protein levels were detected in the clpC1(2) or clpP2(2) knock-down strains but not in the parental bacteria, despite significant induction of hsp20 transcripts. In summary, the cellular levels of oligomeric proteins such as Hsp20 are maintained post-translationally through their recognition, disassembly, and degradation by ClpC1, which requires disordered ends in its protein substrates.”

Considering the results previously reported, it is not at all surprising that in the presence of ecumicin, an inhibitor of ClpC1P1P2 protein degradation, the levels of Hsp20 should increase. Moreover, it also explains the result with ilamycin, that is also an inhibitor of ClpC1P1P2 protein degradation. Cyclomarin, in contrary, has been proposed to enhance protein degradation in vivo: see “Toxic Activation of an AAA+ Protease by the Antibacterial Drug Cyclomarin A” by Maurer et al. However, and despite this previous evidence the authors advance the thesis that since ecumicin also binds to Hsp20, this somehow would protect the Hsp20 chaperone from degradation. Several issues arise here:

- 1) The authors report a MIC90 of 0.15 μM for ecumicin, a concentration that was used in the experiments. With this concentration of ecumicin, and the reported Hsp20 KD (circa 20 μM), the occupancy of Hsp20 with ecumicin would be certainly very low – incompatible with your thesis.
- 2) Ecumicin was recently proposed by the Clausen group to mimic unfolded protein substrates. It is therefore not surprising that it binds to a chaperone that binds unfolded proteins like Hsp20. Did you test other random chaperones? Like Hsp70, HtpG or even ClpB. Since the KD is so high it would not be surprising that some binding can also be observed with other chaperones.
- 3) The authors make a possible analogy between the roles of ClpC2 and Hsp20 but several key differences exist – the most important being the fact that ClpC2 is a very good binder of cyclomarin (KD 2 nM) as described in “ClpC2 protects mycobacteria against a natural antibiotic targeting ClpC1-dependent protein degradation” by Taylor et al.

Major Point 4:

One major problem of using proteomics with the ClpC1P1P2 system is the fact that some of the targets of the protease are themselves cellular regulators. For example ClgR, that is degraded by ClpC1P1P2, is a transcriptional activator that activates the expression of multiple other proteins. It has been shown that ClgR positively regulates several other genes, including its own gene, clgR, as well as *acr2*, encoding a chaperone. This to say, inhibiting ClpC1P1P2, will lead to an increase of ClgR and several other proteins as consequence. So it is not surprising that some protein levels are increased while others are decreased making the very analysis complex. This is not discussed at any point of the article. In our view, to exclude effects like the above, it would be important to test in vitro for degradation with ClpC1P1P2 the substrates with different results between antibiotics, that is - the ones that are specifically increased or decreased with one specific antibiotic.

That way you could test if indeed there is a direct impact of ecumicin, cyclomarin and rufomycin on ClpC1 that makes it more prone for the degradation of some specific substrates? This is a very important question that cannot be answered only using systematic proteomics.

Overall this is an interesting article, but likely more fitting to publications dedicated to proteomics. A broad journal of excellence such as Nature Communications appears perhaps not appropriate unless a detailed mechanistical analysis is performed on the ClpC1P1P2 different substrates.

Reviewer #2

(Remarks to the Author)

Reviewer #3

(Remarks to the Author)

The protein quality control system of mycobacteria is a promising and developing target for mycobacterial drug discovery, which can also be utilized for targeted protein degradation. This manuscript explores the ClpC1:ClpP1P2 protease system in virulent *M. tuberculosis*, focusing on the response to ClpC1 NPA targeted antibiotics - ecumicin, ilamycin, and cyclomarin using a proteomic approach. These studies show key differences in the effects of the antibiotics studied despite their common targeting of ClpC1. The roles of ClpC2 and Hsp20, which help provide compensatory and potentially protective effects to the differing agents, as highlighted in Figure 5, represent the most interesting aspect of this study. Other significant results include studies of the effects of NPA treatment on the abundance of proteins with unstructured domains and termini that are believed to be ClpC1 native substrates, and in the potentiation of the heat shock response.

Overall, this is an excellent proteomics study, in an important topic area on the drug effects of ClpC1 targeted antibiotics. The manuscript is well written and the supporting information adds to the scientific rigor of the reported study.

Primary concerns:

1. Methodological: the proteomic sampling is performed at a single time point after 48h of treatment at the predetermined low OD MIC90 level of the respective drugs. This is too long to study mechanistic effects as compensatory effects will likely dominate the response. Proteomic sampling should be performed in less than one doubling time (24h), and 3-6h is recommended. The authors should also establish that there is not an inoculum effect at the higher cell concentration used in the proteomics experiments that negates the drug action.
2. The second concern relates to the limited scope of the study, which only uses one method of proteomic profiling. Adding complementary unbiased RNAseq or thermal proteome (CETSA) profiling experiments would strengthen the conclusions.
3. Due to its limited scope, this study does not adequately prove the author's conclusion that these antimycobacterial drugs disrupt protein quality control beyond a single protein target.

Other comments and suggestions:

1. The authors use flattering terms like "cutting-edge" and "state of the art" to describe their mass-spectrometry proteomics, which is overstated as these methods are widely employed elsewhere.
2. NPA drug stability after 48h should be confirmed under the incubation conditions.
3. Incorporating a protease ClpP1P2 inhibitor control would be a useful comparator in this study to help decipher the mechanistic effects that are ClpC1-specific.
4. Conclusions summary: given that the three inhibitors lack cross-resistance, is it surprising they have differential impacts?
5. The final stated conclusion "we expect that the findings of this study, combined with more detailed structural data in the future, will greatly facilitate the rational design of next generation potent ClpC1 and PQC-targeting TB drug leads." How? It is unclear to this reviewer how this study would aid further drug development; for instance, have any new vulnerabilities been revealed?
6. Materials and methods – *Mtb* growth conditions: please report oxygen and carbon dioxide levels of culture.
7. Materials and methods: cell viability assay; please report the time, drug concentration, and cellular density at which the *Mtb*-treated samples were prepared.

Reviewer #4

(Remarks to the Author)

Tuberculosis (TB) remains one of the leading causes of death from infection worldwide. The high virulence of *Mycobacterium tuberculosis* (*Mtb*) is partly due to its ability to survive and replicate within the alveolar macrophages, establishing reservoirs of live bacteria that promote the persistence and recurrence of the disease. An estimated one-quarter of the global population harbours a latent TB infection (LTBI)², which, although asymptomatic, represents a reservoir for the potential reactivation and transmission of TB. Additionally, there is an increasing concern regarding the number of the multidrug resistant cases. Bater et al., have tested several antimycobacterial cyclic peptide natural products that bind the ClpC1 chaperone component of the system and claimed that they have employed cutting edge mass-spectrometry-based proteomics to determine the effects of NPAs on virulent *Mtb* H37Rv. The study reports 3, 175 protein IDs of hundreds of proteins were differentially abundant with different NPA treatment. The authors claimed that "this study provides a deeper understanding of the distinct mechanisms of antimycobacterial activity of different NPAs through dysregulation of the ClpC1:ClpP1P2 proteolysis system. However as it stands the manuscript is merely descriptive and the results lacks orthogonal validation.

Major points:

According to the authors, NPAs are known to target a major chaperoning element in the ClpC1-P1P2 protein degradation mechanism. However, there is no evidence in this work showing that NPAs interact with the ClpC1-P1P2 protein system;

rather, the authors refer/rely on changes in protein abundance. The authors must offer supporting evidence that the changes observed in the Mtb proteome are the result of NPA interactions with the ClpC1-P1P2 protein system, rather than a general reaction to a foreign peptide. For this, I recommend that the authors use current techniques like cross-linking mass spectrometry to demonstrate that NPAs bind to the complex. The amount of peptide-protein interaction is then assessed using molecular docking simulation.

The study lacks supportive functional assays that aid in understanding not only the mechanism of action but also the extent of NPA in Mtb viability. For example, it has been demonstrated that CLpC2 protects against CymA-induced toxicity; it would be interesting to know how NPA treatment interferes with such a protective model system or simply compare the presence of NPA to the clpC2 knockout strain.

Version 1:

Reviewer comments:

Reviewer #1

(Remarks to the Author)

The authors have clearly made a significant effort to improve the manuscript, and there is no question about their commitment to addressing the previous concerns. However, we still find that the overall message of the paper is not sufficiently strong for publication in Nature Communications, although we believe this is an interesting topic of research.

That said, we also believe it is not the reviewer's role to impose unrealistic demands on a complex topic, particularly when doing so could unnecessarily impact the careers of postdoctoral researchers or PhD students involved in the work – and we are aware it was a massive amount of work. In our view, the study appears to have been carefully executed, but the data are inherently complex, making it difficult to draw clear and definitive conclusions.

Overall, we acknowledge the authors' efforts to improve the article — to do better science — which, in our view, should be the main goal of any review process. Below, we outline several points that we believe will improve the manuscript if addressed before publication:

The Kd of the Ecu*, corrected by the authors here, has been also reported by the Clausen group to be 8 μ M (one-to-one stoichiometry) using also SPR (see Hoi Cell 2023) - Dr. Richard Payne, the senior author, is also an author in that publication. Should we assume the data there was incorrect - is that true? Maybe you should refer to this fact somewhere in the manuscript as other researchers will be misguided by the conflicting results between publications.

Page 3

The authors describe the ClpC1P2P1 system but rely on general references that are not specific to this complex. In addition, several recent structural studies on the ClpC1P2P1 system are omitted – including structures of ClpP1P2, ClpC1, and two recent publications on the ClpC1P2P1 complex. These should be properly cited and discussed.

Please verify whether reference 15 is correct and appropriate in the context where ClpP1 and ClpP2 association for protein hydrolysis is discussed. It would be more relevant to cite the ClpP1P2 structural studies from the Sauer and Goldberg groups.

Page 4, line 76

The authors should mention lassomycin, as it is another well-characterized ClpC1-NTD binding natural product.

Page 5 line 124

dCym was not first described by Hoy et al. – but by Barbie, P. and Kazmaier in 2016.

Mechanistic discussion

There are fundamental differences in the in vitro mechanisms of ecumicin, cyclomarin, and rufomycin when tested with purified proteins. Notably, cyclomarin has not been described as a strong ClpC1P2P1 inhibitor; in fact, prior studies suggest that it activates the ClpC1P1P2 complex (see Mogk Heidelberg group on cyclomarin publications). This distinction should be discussed.

Page 8, line 209

The statement that IDPs are structurally similar to unfolded substrates is inaccurate. IDPs are typically stable and functional, whereas unfolded substrates are often prone to aggregation. This section should be clarified to avoid conceptual confusion.

Page 8, line 232

The manuscript states that bortezomib has no effect on in vitro degradation of casein by the ClpC1P2P1 complex, using a concentration of 100 μ M. This seems inconsistent and potentially confusing. 100 μ M is an extremely high concentration; please justify this choice and, by the way, the reference cited shows that bortezomib activates protein degradation at lower concentrations than 100 μ M. Of note, the effect of the active site activators is also well described in the literature as a bell-shaped curve – that is, activation at low concentrations but inhibition at higher concentrations – please use the citations in the proper context.

In addition, why was 100 μ M used in vitro while 5 μ M? MIC50 or MIC90 was used in vitro? Normally, in vitro assays require lower concentrations to achieve target engagement. Related to this, please clarify what were the concentrations used in the proteomics experiments? Why put the bortezomib MIC50 in the figure when you used the MIC90 in the experiment (was 100 μ M the MIC90?)

Additionally, how were the in vitro degradation experiments performed without a ClpP1P2 activator? You observed activity in the absence of any ClpP1P2 activator? It has been reported by several groups that the ClpP1P2 is not active in the absence of active site activators. Can you explain this?

Page 10, line 281

The authors state that Hsp20 is specifically enriched by NRP action, yet a similar enrichment is observed with bortezomib – a compound that does not bind to ClpC1 and preferentially targets the *M. tuberculosis* proteasome rather than ClpP1P2. This raises concerns about the specificity of the reported effect. The data appear to contradict the claim of selective enrichment – likely any stress in protein quality control will lead to an Hsp20 increase.

Of note, the authors choose not to test the binding of the compounds to other chaperones as we suggested before, we do understand that – but we do believe you would likely get binding as well – but we agree this is not a key issue for review.

Heat stress section

The discussion of heat stress is unclear. Are the authors suggesting that heat stress potentiates the effects of drugs acting on protein quality control? If so, this concept is already well established in the literature and does not constitute a novel finding. The claims in this section should be moderated accordingly, as the data is largely confirmatory.

Reviewer #2

(Remarks to the Author)

Reviewer #3

(Remarks to the Author)

The authors have done an excellent job responding to the prior critique. No further changes are suggested.

Reviewer #4

(Remarks to the Author)

The authors have addressed my previous comments and have now submitted an improved version of the manuscript.

Open Access This Peer Review File is licensed under a Creative Commons Attribution 4.0 International License, which permits use, sharing, adaptation, distribution and reproduction in any medium or format, as long as you give appropriate credit to the original author(s) and the source, provide a link to the Creative Commons license, and indicate if changes were

made.

Responses to Referees:

Referee #1:

Minor Point

There are multiple previous articles using the terminology rufomycin instead of ilamycin. It is true that the authors explain that ilamycins are the same as rufomycins, but perhaps this should be done earlier in the text.

We thank the reviewer for this comment. We have moved this clarification to the abstract (page 2, line 8) and also reiterated in the main text (page 4, line 76).

Point 1:

The authors state: “Ecu*(1) bound to MtbClpC1NTD with a KD of 8.58 μ M, IlaE(2) a KD of 0.11 μ M and dCym(3) a KD of 0.013 μ M. These values are consistent with those reported in previous studies 41.”

We believe that the reported values for ecumicin and rufomycin are significantly lower as for example in the following article (that is not cited here), “High-Resolution Structure of ClpC1-Rufomycin and Ligand Binding Studies Provide a Framework to Design and Optimize Anti-Tuberculosis Leads” by Wolf et al, the values described for ecumicin and rufomycin (also using SPR) are very different. For ecumicin binding to ClpC1-NTD the KD value was described as 50 nM and for rufomycin the value was 63 nM. The KD for ecumicin in that previous publication is therefore circa 200x lower than the one you describe here and that is even more puzzling when you state here the following: “a synthetic ecumicin analogue (Ecu*, 1) with a simplified amino acid composition, and superior potency, to the parent natural product”.

So either your ecumicin analog is in fact less potent than you propose or the KD value you present here is not correct. This paradox is even more puzzling when you report a MIC90 of 0.15 μ M for ecumicin (supplementary Fig1) which was used in the proteomics experiments. Of note, if the ecumicin analog is not as potent as you propose, it would be perhaps important to do this kind of studies with the more potent compound.

We thank the reviewer for their comment. We do agree that our reported KD for Ecu* was low and, after further consulting the published literature, have remeasured SPR data with a revised protocol. Specifically, we have repeated the SPR experiments at a more suitable concentration range and fit the sensorgram to multi-site parameters, in alignment with the reported 2:1 binding mode of the Ecu natural product to ClpC1 (*Acta Crystallogr D Struct Biol*, 2020, 458). We have recalculated the affinity of Ecu* to ClpC1 as KD1 = 0.39 μ M and KD2 = 1.7 μ M. These changes have been updated in Figure 4c, Supplementary Figure 9a and associated text.

There is precedent that subtle changes in the structure of ecumicin can result in changes to both binding affinity to ClpC1-NTD measured by SPR and the MIC against Mtb. For example, ohmyungsamycin A (OMSA) is a related congener of ecumicin and exhibits weaker binding to ClpC1 by SPR ($K_{D1} = 135 \text{ nM}$ vs 22.5 nM (*Acta Crystallogr D Struct Biol*, 2020, 458) but exhibits more potent activity against Mtb ($MIC_{90} = 110 \text{ nM}$ vs 360 nM for the parent ecumicin natural product, *Chem Eur J*, 2020, 15200). It would therefore be reasonable to observe a discrepancy between the binding affinity of the Ecu natural product and analogue Ecu* to the immobilised monomeric fragment of ClpC1 and the activity against the target microorganism, where permeability, intracellular stability, and the interaction with a hexameric ClpC1 in the ClpC1P1P2 complex are all important considerations. With respect to ilamycin E, the K_D reported by Wolf *et al.* is for rufomycin 4 (ilamycin C) which is a different compound to the one we have synthesized and analysed (ilamycin E). Nonetheless, the binding affinity measured for Ruf4 against ClpC1 ($K_D = 63 \text{ nM}$) is similar to that measured for ilamycin E in our study ($K_D = 107 \text{ nM}$).

New sensorgrams and affinity data are included in the revised Figure 4c and Supplementary Fig 7 and are also provided below:

NRP	K_D (μM)	
	ClpC1-NTD	ClpC2-CTD
Ecu* (1)	$K_{D1} = 0.39 \pm 0.12$ $K_{D2} = 1.7 \pm 0.9$	43.8 ± 2.2
IlaE (2)	0.107 ± 0.024	0.69 ± 0.06
dCym (3)	0.013 ± 0.001	0.026 ± 0.06

Figure 4c. Surface plasmon resonance (SPR) binding affinities for interaction between NRPs and ClpC1-NTD and ClpC2-CTD. Raw data for three replicates is presented in Supplementary Figure 7.

Supplementary Figure. 9| Raw data for surface plasmon resonance (SPR) binding assays shows each NRP bound another chaperone in addition to ClpC1. a SPR plots for all replicates of NRPs 1-3 ligand binding with recombinant Mtb ClpC1-NTD replicates. **b** SPR plots for all replicates of NRPs 1-3 interaction with recombinant Mtb ClpC2-CTD. **c** SPR plots for NRP ligand binding to recombinant Mtb Hsp20.

Point 2:

“Increased temperature has been shown to increase the bactericidal effect of cyclomarlin; this is thought to be due to occupation of ClpC2 by unfolded proteins, thus allowing cyclomarlin to act on ClpC1 41.”

Another reason for this high temperature effect, which is more likely in our view, is that high

temperature increases the burden of unfolded proteins, challenging the protein quality system and therefore making the cell more sensitive to its dysregulation by antibiotics. To test the author's hypothesis, it would be important to inhibit just ClpP1P2 (for example with bortezomib), this way bypassing ClpC1, and check if you would obtain similar results with temperature increase. If not, your hypothesis would be supported.

We thank the reviewer for this comment and suggested experiment. The effect of cyclomarin under heat stress described in the manuscript is based on conclusions drawn from prior work by Hoi *et al.* (Cell 2023, 2176). Nonetheless, in new experiments, we inhibited Mtb proteases with bortezomib (BTZ) to bypass ClpC1 and assessed Mtb survival under mild heat stress. BTZ treatment in combination with elevated temperatures caused a significant drop in viability (Figure 5h), consistent with heat increasing the burden of unfolded proteins and sensitizing cells to protease inhibition. The new data can be found in Figure 5h of the revised manuscript, and these new results are described on page 11, from line 322.

Fig. 5 | h Colony forming unit (CFU) viability assay for heat stressed and non-heat stressed *Mtb* H37Rv treated with NRPs 1-3 and BTZ (4). Data presented as mean \pm SD, n = 3 for each condition. Statistical significance was calculated with a 2-way ANOVA, * = p-value < 0.05, ** = p-value < 0.01. Source data are provided in Source data file.

Point 3:

The authors cite and refer multiple times to the article “The unfoldase ClpC1 of *Mycobacterium tuberculosis* regulates the expression of a distinct subset of proteins having intrinsically disordered termini” by Lunge *et al.* In this specific publication the authors show that ClpC1 is responsible for the degradation of Hsp20, it is clearly stated:

“Further, we observed that Hsp20 is poorly expressed in WT *Mtb* and that its expression is greatly induced upon depletion of clpC1 or clpP2. Remarkably, high Hsp20 protein levels were detected in the clpC1(2) or clpP2(2) knock- down strains but not in the parental bacteria, despite significant induction of hsp20 transcripts. In summary, the cellular levels of oligomeric proteins such as Hsp20 are maintained post-translationally through their recognition, disassembly, and degradation by ClpC1, which requires disordered ends in its protein substrates.”

Considering the results previously reported, it is not at all surprising that in the presence of ecumicin, an inhibitor of ClpC1P1P2 protein degradation, the levels of Hsp20 should increase. Moreover, it also explains the result with ilamycin, that is also an inhibitor of ClpC1P1P2 protein degradation. Cyclomarin, in contrary, has been proposed to enhance protein degradation in vivo: see “Toxic Activation of an AAA+ Protease by the Antibacterial Drug

Cyclomarlin A” by Maurer et al. However, and despite this previous evidence, the authors advance the thesis that since ecumicin also binds to Hsp20, this somehow would protect the Hsp20 chaperone from degradation. Several issues arise here:

1) The authors report a MIC90 of 0.15 μM for ecumicin, a concentration that was used in the experiments. With this concentration of ecumicin, and the reported Hsp20 KD (circa 20 μM), the occupancy of Hsp20 with ecumicin would be certainly very low – incompatible with your thesis.

We thank the reviewer for these comments. Upon revision of the manuscript (including support from new experiments suggested by all reviewers) we agree that our earlier thesis that Hsp20 may sequester Ecu* was likely incorrect. As part of our revision we repeated SPR experiments across a more suitable concentration range and recalculated the affinity of Ecu* for Hsp20 to be 3.5 μM (Figure 5c, Supplementary Figure 9c, line 299 page 10). While this represents an improvement in affinity, the interaction remains markedly weaker than that of Ecu* and ClpC1, making an Hsp20 sequestration mechanism unlikely. We have revised the manuscript text to reflect these data in the results and explicitly in the discussion (line 425, page 14).

2) Ecumicin was recently proposed by the Clausen group to mimick unfolded protein substrates. It is therefore not surprising that it binds to a chaperone that binds unfolded proteins like Hsp20. Did you test other random chaperones? Like Hsp70, HtpG or even ClpB. Since the KD is so high it would not be surprising that some binding can also be observed with other chaperones.

We agree that as the non-ribosomal peptides (NRPs) are proposed to mimic unfolded proteins, interactions with other chaperones or protein quality control proteins are possible. However, our proteomic analysis shows that each of the NRPs lead to strong dysregulation of Hsp20 and ClpC2, whereas levels of other major chaperones including Hsp70, HtpG and ClpB remained largely unaffected (see Supplementary Figure 8 and Figure 1e-g). This striking response guided our focus on Hsp20 and ClpC2, supporting their functional relevance in our system. We believe that investigation into other chaperones is out of the scope of our already comprehensive study.

3) The authors make a possible analogy between the roles of ClpC2 and Hsp20 but several key differences exist – the most important being the fact that ClpC2 is a very good binder of cyclomarlin (KD 2 nM) as described in “ClpC2 protects mycobacteria against a natural antibiotic targeting ClpC1-dependent protein degradation” by Taylor *et al.*

We agree with the reviewer that Hsp20 and ClpC2 are not directly comparable; Mtb ClpC2 binds dCym with high affinity (KD 13 nM), providing potential for a clear protective mechanism, whereas the weaker interaction between Ecu* and Hsp20 makes efficient sequestration less likely. As described for the comment from the same reviewer above, we have updated the text in the discussion of the revised manuscript to clarify this point.

Point 4:

One major problem of using proteomics with the ClpC1P1P2 system is the fact that some of the targets of the protease are themselves cellular regulators. For example ClgR, that is degraded by ClpC1P1P2, is a transcriptional activator that activates the expression of multiple other proteins. It has been shown that ClgR positively regulates several other genes, including

its own gene, *clgR*, as well as *acr2*, encoding a chaperone. This to say, inhibiting ClpC1P1P2, will lead to an increase of ClgR and several other proteins as consequence. So it is not surprising that some protein levels are increased while others are decreased making the very analysis complex. This is not discussed at any point of the article.

In our view, to exclude effects like the above, it would be important to test *in vitro* for degradation with ClpC1P1P2 the substrates with different results between antibiotics, that is - the ones that are specifically increased or decreased with one specific antibiotic. That way you could test if indeed there is a direct impact of ecumicin, cyclomarin and rufomycin on ClpC1 that makes it more prone for the degradation of some specific substrates? This is a very important question that cannot be answered only using systematic proteomics.

We thank the reviewer for highlighting the complexities of interpreting proteomic changes in response to dysregulation of the ClpC1P1P2 system. To address this, we have now assessed the activity of each of the NRPs on *in vitro* the degradation of model substrate β -casein and native Mtb ClpC1P1P2 substrates, PanD and Hsp20, by reconstituted ClpC1P1P2. The new data can be found in Figure 3, Supplementary Figure 6 in the revised submission and below for convenience. It is worth noting that all previous studies on Mtb ClpC1P1P2 have assessed activity on exclusively model substrates and this data represents the first direct evidence that the NRPs may affect ClpC1P1P2 activity on specific substrates in a NRP-dependent manner. In our hands, ClgR could not be generated with the purity necessary to use in a degradation assay. The text associated with this new data can be found under *Perturbation of ClpC1P1P2 substrate degradation by NRPs*, (page 8, line 227).

In addition to these experiments, on the suggestion of reviewer 3, we carried out transcriptomic analysis on Ecu* treated Mtb to distinguish between changes due to dysregulation at the protein and transcript level. We particularly focused on ClgR and the proteins it regulates including Hsp20, ClpC1, ClpP1 and ClpP2. The new data and discussion for these experiments can be found in Figure 5 of the revised manuscript and described from line 307, page 11.

Figure 3 | d *In vitro* ClpC1P2 degradation assays with substrates; β -casein, PanD and Hsp20 in the presence of 100 μ M NRP. Timepoints taken at 0, 10, 20 and 60 min for β -casein and PanD, 0, 60, 120 and 180 min for Hsp20. **e** Quantification of *in-vitro* degradation assays. Data is presented as mean \pm SD, $n \geq 3$. Statistical significance between endpoints was calculated using two-tailed unpaired *t*-tests, * = p -value < 0.05, ** = p -value < 0.01, *** = p -value < 0.001, **** = p -value = < 0.0001. **f** Analytical size exclusion chromatography depicting oligomeric state of Hsp20 under degradation assay conditions. Source data are provided in Source data file.

Overall this is an interesting article, but likely more fitting to publications dedicated to proteomics. A broad journal of excellence such as Nature Communications appears perhaps not appropriate unless a detailed mechanistical analysis is performed on the ClpC1P2 different substrates.

We would like to express our great appreciation for the reviewer's careful assessment and feedback on our manuscript. We believe the additional experiments we have performed on the suggestion of the reviewer have greatly improved the impact of our work, particularly surrounding the influence of NRPs on the degradation on specific substrates; our paper now describes the first report of the effect of NRPs on bona fide Mtb ClpC1 substrates. We hope that with these experiments and those suggested by reviewers 2-4, the reviewer is satisfied with the importance of our work and its appropriateness for publication in *Nature Communications*.

Reviewer #2 (Remarks to the Author)

I co-reviewed this manuscript with one of the reviewers who provided the listed reports. This

is part of the Nature Communications initiative to facilitate training in peer review and to provide appropriate recognition for Early Career Researchers who co-review manuscripts.

Reviewer #3 (Remarks to the Author)

The protein quality control system of mycobacteria is a promising and developing target for mycobacterial drug discovery, which can also be utilized for targeted protein degradation. This manuscript explores the ClpC1:ClpP1P2 protease system in virulent *M. tuberculosis*, focusing on the response to ClpC1 NPA targeted antibiotics - ecumicin, ilamycin, and cyclomarin using a proteomic approach. These studies show key differences in the effects of the antibiotics studied despite their common targeting of ClpC1. The roles of ClpC2 and Hsp20, which help provide compensatory and potentially protective effects to the differing agents, as highlighted in Figure 5, represent the most interesting aspect of this study. Other significant results include studies of the effects of NPA treatment on the abundance of proteins with unstructured domains and termini that are believed to be ClpC1 native substrates, and in the potentiation of the heat shock response.

Overall, this is an excellent proteomics study, in an important topic area on the drug effects of ClpC1 targeted antibiotics. The manuscript is well written and the supporting information adds to the scientific rigor of the reported study.

We thank the reviewers for their thoughtful and encouraging comments. We are grateful for their recognition of the importance our study, particularly regarding the differential effects of CpC1-targeted NPAs, the roles of ClpC2 and Hsp20, and the insights provided by our proteomic analyses.

Primary concerns:

1. Methodological: the proteomic sampling is performed at a single time point after 48h of treatment at the predetermined low OD MIC90 level of the respective drugs. This is too long to study mechanistic effects as compensatory effects will likely dominate the response. Proteomic sampling should be performed in less than one doubling time (24h), and 3-6h is recommended. The authors should also establish that there is not an inoculum effect at the higher cell concentration used in the proteomics experiments that negates the drug action.

We thank the reviewer(s) for this thoughtful comment. We selected 48 hours to capture robust, proteome-wide changes and maintain consistency with the literature that had used the same time point for cyclomarin studies (*Cell*, 2023, 2176). We initially assessed the impact of Ecu* on the Mtb proteome at 16 hours (less than one doubling time) and observed minimal changes to the proteome. We have now included these data in Supplementary Figure 2 of the revised submission and inserted below. This aligns with previous reports indicating that effects become pronounced only after one or more doubling times (*J Med Chem*, 2022, 4893; *Tuberculosis*, 2025, 102594; *Cell*, 2023, 2176).

Regarding the inoculum effect, which refers to the reduction in apparent antibiotic efficacy at higher bacterial densities, we observed no increase in optical density between 16 and 48 hours (data now presented in Supplementary Figure 2 of the revised submission and inserted below), indicating that the effective drug to cell ratio is maintained and that classical inoculum effects are unlikely to influence the observed proteomic changes.

Supplementary Fig. 2 | a Comparison between LC-MS/MS based proteomic analysis of Mtb H37Rv treated with MIC₉₀ of **1** following a 16 h or a 48 h incubation, n ≥ 3. **b** change in optical density at 600 nm following treatment at 16 h and 48 h. Data is presented as mean ± SEM, n ≥ 3. **c** Colony forming units (CFU) Mtb H37Rv cell viability assay was performed to determine lethality of a treatment with the MIC₉₀ of each of the NRPs **1-3** for 48 h. Data is presented as mean ± SEM, n = 3.

2. The second concern relates to the limited scope of the study, which only uses one method of proteomic profiling. Adding complementary unbiased RNAseq or thermal proteome (CETSA) profiling experiments would strengthen the conclusions.

We thank the reviewer for this suggestion and agree that a complementary technique would provide orthogonal validation of our proteomics data. We have now performed transcriptomic profiling following Ecu* treatment. Substantial remodeling of the transcriptome was observed, with enriched proteins showing greater overlap with upregulated transcripts, indicating that protein depletion is largely post-translational, while enrichment reflects both transcriptional activation and reduced turnover. Notably PQC related genes, including Hsp20, ClgR, ClpC1, ClpP1 and ClpP2 were significantly upregulated at the transcript level, and this may explain the mild enrichment of ClpC1, ClpP1 and ClpP2 in the proteomic dataset. The data for these experiments can be found in Figure 5 (inserted below) and are described from line 307, page 11. These results reinforce the added value of integrating RNA-seq with proteomics to dissect the complex response of ClpC1 dysregulating NRPs.

Figure 5 | **e** Transcriptomic analysis of Ecu* (1) treated Mtb H37Rv following 48 h incubation. Red dots represent downregulated transcripts ($< -1 \log_2$ fold change, p -value < 0.05). Blue dots represent upregulated transcripts ($> 1 \log_2$ fold change, p -value < 0.05). Clp proteins and highly upregulated Hsp20 are labelled. **f** Venn diagrams showing number of similarly or uniquely affected proteins across transcriptomic and proteomic datasets for Ecu* (1) treated Mtb, upregulated (top) or downregulated (bottom). Enriched or depleted transcripts/ proteins were those significantly dysregulated greater than ± 1 -fold. **g** Extracted normalised counts for the transcripts for specific PQC proteins. Data presented as mean \pm SD, $n = 4$ for each condition and data was analysed by two-tailed Wald test, **** = p -value < 0.0001 . Source data are provided in Supplementary Data file 5.

3. Due to its limited scope, this study does not adequately prove the author's conclusion that these antimycobacterial drugs disrupt protein quality control beyond a single protein target.

We thank the reviewer for this comment. Based on the additional experiments suggested by all reviewers, including degradation assays with Mtb client proteins, RNA sequencing and Mtb knockdown strains, we have refined the key conclusions of the manuscript and have adjusted the discussion and title of the manuscript accordingly. While the NRPs clearly modulate ClpC1/P2 activity, we now place less emphasis on broad, generalized disruption of protein quality control and instead highlight substrate and compound specific effects supported by both *in vitro* and *in cell* data.

Other comments and suggestions:

1. The authors use flattering terms like “cutting-edge” and “state of the art” to describe their mass-spectrometry proteomics, which is overstated as these methods are widely employed elsewhere.

We have adjusted the language accordingly throughout the manuscript.

2. NPA drug stability after 48h should be confirmed under the incubation conditions.

We have now confirmed the stability of each of the NRPs under incubation conditions, and the experimental data can be found in Supplementary Figure 2d in the revised submission and included below:

d Stability of Ecu* (1) under incubation conditions

Stability of IlaE (2) under incubation conditions

Stability of dCym (3) under incubation conditions

Supplementary Figure 2 | d Stability of Ecu* (1, 1.0 μ M), IlaE (2, 10 μ M), dCym (3, 1.0 μ M) in Middlebrook 7H9 media containing 0.2% glycerol at 37 °C. Samples were analysed directly by UPLC-MS (0-100% MeCN in H₂O over 5 min with 0.1 vol.% FA). *a* and *b* denote *S* and *R* epimers of the 6-hydroxypiperidin-2-one motif, which readily interconvert in aqueous conditions.

3. Incorporating a protease ClpP1P2 inhibitor control would be a useful comparator in this study to help decipher the mechanistic effects that are ClpC1-specific.

This is a great suggestion. We have now performed LC MS/MS based proteomics on *Mtb* H37Rv treated with well documented protease inhibitor bortezomib (BTZ). As this experiment had not been performed previously in the literature, we chose to analyze the effects BTZ on the proteome of *Mtb* at two timepoints, 16 h and 48 h and compared both to the NRP datasets. Interestingly, BTZ treatment led to dramatic effects on the *Mtb* proteome, but these were quite different to the changes induced by the ClpC1-targeting NRPs. The data for this experiment can be found in Figure 2 (provided below), Supplementary Figures 4, 5 and 9) in the revised submission. The description of the results can be found under *BTZ inhibits protein degradation in Mtb*, line 167, page 6 and within the discussion section.

Figure 2 | Bortezomib inhibits protein degradation in Mtb H37Rv. a Structure of Bortezomib (BTZ, 4). Minimum inhibitory concentration (MIC) to inhibit 50% of bacteria (MIC₅₀) indicated below structure. Raw data from the resazurin MIC assay found in Supplementary Fig. 1d and source data provided in Source data file. **b** Schematic of ClpC1P2 complex with compound target site indicated, NRPs bind to the NTD of ClpC1 and BTZ (4) interacts with the protease. **c** Volcano plots representing the effects of BTZ (4) on the proteome of Mtb H37Rv (top = 16 h incubation, bottom = 48 h incubation). Statistical significance was calculated with a two-way ANOVA (* = p-value < 0.05). Red dots represent depleted proteins (< -0.5 log₂ fold change, p-value < 0.05). Blue dots represent enriched proteins (> 0.5 log₂ fold change, p-value < 0.05). Clp proteins and highly upregulated Hsp20 and ClgR are labelled. 16 h dataset includes n = 4 biological replicates and 48 h includes n = 3 biological replicates. **d, e** Venn diagrams showing number of similarly or uniquely affected proteins, depleted (left) or enriched (right)

between NRPs (1-3) and BTZ (4) 16 h (d) and BTZ (4) 48 h (e). **f, g** Raw abundance of ClpC1 substrates (f) and 20S proteasome substrates (g) following BTZ (4) treatment. Data is presented as mean \pm SEM, 16 h n = 4, 48 h n = 3, * = p-value < 0.05, ** = p-value < 0.01, *** = p-value < 0.001, **** = p-value < 0.0001. Source Data are provided in Supplementary Data 4.

4. Conclusions summary: given that the three inhibitors lack cross-resistance, is it surprising they have differential impacts?

We thank the reviewer for this comment. We believed it was important to comment that subtle differences in the precise positioning and contacts can markedly influence how ClpC1 engages its substrates, highlighting that even closely related compounds can exert distinct effects through the same binding interface.

The statement in our originally submitted manuscript was incorrect; mutations at F80 and V13 confer resistance to more than one natural product (see Supplementary Figure 1a). We have amended this in the revised manuscript to:

“There is considerable ClpC1-interface overlap between the three compounds and mutagenesis studies have revealed mutations at F80 and V13 confer resistance to more than one natural product. (Figure 1b, Supplementary Figure 1a).”

5. The final stated conclusion “we expect that the findings of this study, combined with more detailed structural data in the future, will greatly facilitate the rational design of next generation potent ClpC1 and PQC-targeting TB drug leads.” How? It is unclear to this reviewer how this study would aid further drug development; for instance, have any new vulnerabilities been revealed?

We would like to thank the reviewer for raising this. As a result of the new data that we have included based on the suggestions from the four reviewers we have added additional clarification to our concluding paragraph. Specifically, based on our data it is unlikely that Ecu* and IlaE are sequestered by ClpC2 (unlike dCym). This reveals that not all NRPs are susceptible to inbuilt Mtb protective mechanisms and positions ecumicin and ilamycins as promising molecular scaffolds for rational design of ClpC1-targeting drug candidates, including BacPROTAC molecules. The new sentence in the concluding paragraph is provided below:

“Indeed, we show that Ecu (1) and IlaE (2) cause extensive proteome dysregulation at sublethal doses but do not engage the ClpC2 rescue mechanism that reduces dCym (3) toxicity. The ability of ecumicin and the ilamycins to overcome ClpC2 rescue, that would serve to reduce their antimycobacterial activity, makes these molecules promising molecular scaffolds for rational drug design of ClpC1-targeting ligands and as ClpC1 recruiters for next-generation BacPROTACs. Taken together, the distinct interactions, combined with the differential dysregulation of ClpC1/P2 proteolysis, highlight the multifaceted ways NRPs perturb proteostasis. Given the essentiality of ClpC1 and protein quality control in Mtb, these findings reinforce the potential to exploit stress-adaptive pathways to enhance antimycobacterial activity and provide the foundation for the rational design of next-generation ClpC1- and PQC-targeting TB drug candidates.”*

6. Materials and methods – Mtb growth conditions: please report oxygen and carbon dioxide levels of culture.

The oxygen and carbon dioxide levels for Mtb culturing (20% and 5%, respectively) have been included and highlighted in the Materials and Methods section of the manuscript (line 461, Page 15)

7. Materials and methods: cell viability assay; please report the time, drug concentration, and cellular density at which the Mtb-treated samples were prepared.

We have now included these important details in the Methods of the revised submission (line 521, page 17).

Reviewer #4 (Remarks to the Author):

Tuberculosis (TB) remains one of the leading causes of death from infection worldwide. The high virulence of Mycobacterium tuberculosis (Mtb) is partly due to its ability to survive and replicate within the alveolar macrophages, establishing reservoirs of live bacteria that promote the persistence and recurrence of the disease. An estimated one-quarter of the global population harbours a latent TB infection (LTBI)², which, although asymptomatic, represents a reservoir for the potential reactivation and transmission of TB. Additionally, there is an increasing concern regarding the number of the multidrug resistant cases. Bater et al., have tested several antimycobacterial cyclic peptide natural products that bind the ClpC1 chaperone component of the system and claimed that they have employed cutting edge mass-spectrometry-based proteomics to determine the effects of NPAs on virulent Mtb H37Rv. The study reports 3,175 protein IDs of hundreds of proteins were differentially abundant with different NPA treatment. The authors claimed that “this study provides a deeper understanding of the distinct mechanisms of antimycobacterial activity of different NPAs through dysregulation of the ClpC1:ClpP2 proteolysis system. However as it stands the manuscript is merely descriptive and the results lack orthogonal validation.

While our original manuscript relied primarily on proteomic profiling, we have now incorporated multiple orthogonal approaches to strengthen mechanistic insights, including ClpC1 degradation assays with client proteins, RNAseq analysis and comparisons of the effects of the NPAs on the Mtb proteome with the protease inhibitor BTZ. Collectively these experiments provide evidence that the observed effects of NPAs on ClpC1 are substrate- and compound-specific. We have amended the text in the revised manuscript to include these validation experiments meaning that our discussion is now less ‘descriptive’ than the original submission.

Point 1:

According to the authors, NPAs are known to target a major chaperoning element in the ClpC1-ClpP2 protein degradation mechanism. However, there is no evidence in this work showing that NPAs interact with the ClpC1-ClpP2 protein system; rather, the authors refer/rely on changes in protein abundance. The authors must offer supporting evidence that the changes observed in the Mtb proteome are the result of NPA interactions with the ClpC1-ClpP2 protein system, rather than a general reaction to a foreign peptide. For this, I recommend that the authors use current techniques like cross-linking mass spectrometry to demonstrate that NPAs bind to the complex. The amount of peptide-protein interaction is then assessed using molecular docking simulation.

To directly link the observed proteomic changes to ClpC1P1P2 activity, we performed *in vitro* degradation assays using model substrate β -casein and genuine Mtb ClpC1P1P2 substrates PanD and Hsp20. These assays demonstrate that NRPs differentially modulate ClpC1P1P2 dependent proteolysis in a substrate-specific manner (line 227, page 8). In addition, we used bortezomib as a comparator to distinguish ClpC1-specific effects, further supporting that the proteomic changes observed in cells arise from targeted modulation of ClpC1P1P2 system rather than inhibition of other cellular pathways (line 167, page 6).

The interaction of the NRPs with ClpC1 is also well established in the literature through biochemical, biophysical and structural studies (*Acta Crystallogr D Struct Biol*, 2020, 458; *ACS Infect Dis*, 2019, 829; *Cell*, 2023, 2176; *Antimicrob Agents Chemother*, 2019, e02317; *Cell Chem Biol*, 2019, 1169; *Commun Biol*, 2023, 301), reinforcing that ClpC1 targeting is the primary mechanism of action.

The study lacks supportive functional assays that aid in understanding not only the mechanism of action but also the extent of NPA in Mtb viability. For example, it has been demonstrated that CLpC2 protects against CymA-induced toxicity; it would be interesting to know how NPA treatment interferes with such a protective model system or simply compare the presence of NPA to the clpC2 knockout strain.

We thank the reviewer for this great suggestion. As part of this revision we generated a ClpC2 knockdown strain of Mtb H37Rv and assessed susceptibility to each NPA. Consistent with previous studies, susceptibility to dCym was significantly increased in the ClpC2 knockdown. Notably, this is the first report testing Ecu* and IlaE in the context of ClpC2 depletion, and we found their activity was unaffected, consistent with the lack of ClpC2 enrichment observed in our proteomic analysis. These data are now presented in Figure 4 of the revised manuscript (see below) and associated text can be found under *compound specific modulation of ClpC2 and impacts on NRP toxicity*, line 259, page 9.

Figure 4| e Relative mRNA levels of ClpC2 knockdown in Mtb mc²6026 determined by qPCR and protein abundance determined by LC-MS/MS based proteomics. The control strain contained non-targeting sgRNA. Data is presented as mean \pm SD and each dot represents an individual biological replicate, n = 3. Statistical significance was calculated with two-tailed unpaired *t*-tests, ** = p-value <

0.01. **f** Growth inhibition curves from resazurin cell viability assay comparing NRP activity in control and ClpC2 knockdown *Mtb mc²6026* strains. Data presented as mean \pm SD from one biological replicate with three technical replicates. **g** IC₅₀ quantification from resazurin cell viability assays. Data is presented as mean \pm SEM, dots represent data from three individual biological replicates, $n \geq 3$ and analysed by two-tailed unpaired *t*-test on log-transformed data. Source data for all replicates are provided in Source data file.

Reviewer #1 (Remarks to the Author)

The authors have clearly made a significant effort to improve the manuscript, and there is no question about their commitment to addressing the previous concerns. However, we still find that the overall message of the paper is not sufficiently strong for publication in *Nature Communications*, although we believe this is an interesting topic of research.

That said, we also believe it is not the reviewer's role to impose unrealistic demands on a complex topic, particularly when doing so could unnecessarily impact the careers of postdoctoral researchers or PhD students involved in the work – and we are aware it was a massive amount of work. In our view, the study appears to have been carefully executed, but the data are inherently complex, making it difficult to draw clear and definitive conclusions.

Overall, we acknowledge the authors' efforts to improve the article — to do better science — which, in our view, should be the main goal of any review process. Below, we outline several points that we believe will improve the manuscript if addressed before publication:

We thank the reviewer for recognizing the effort we have invested in addressing the previous comments in our revised the manuscript and that the work has been carefully executed.

While we agree that the area of mycobacterial protein degradation is complex, our work uncovers several pivotal discoveries that open new research pathways in mycobacterial biology and provides a foundation for new TB drug discovery efforts. Below we have summarised four key findings as exemplars to emphasise the importance and impact of our work:

1. ClpC1-binding natural products exert distinct effects on the proteome of virulent Mtb, despite overlapping binding interfaces.
2. We demonstrate that, unlike dCym, Ecu* and IlaE do not trigger the ClpC2-mediated ClpC1-rescue mechanism. This was confirmed through binding studies, proteomic profiling, and inhibitory activity evaluation in a Mtb ClpC2 knockdown strain.
3. We show that Ecu* and IlaE each trigger an increase in small heat shock protein Hsp20 in Mtb cells (32-95-fold increase over controls). While bortezomib also triggers Hsp20 production, this molecule broadly inhibits protein degradation in Mtb cells and triggers a complete, coordinated protein quality control response.
4. We demonstrate that ecumicin binds to Hsp20, a non-Clp chaperone, in addition to ClpC1.

The K_D of the Ecu*, corrected by the authors here, has been also reported by the Clausen group to be 8 μ M (one-to-one stoichiometry) using also SPR (see Hoi *Cell* 2023) - Dr. Richard Payne, the senior author, is also an author in that publication. Should we assume the data there was incorrect - is that true? Maybe you should refer to this fact somewhere in the manuscript as other researchers will be misguided by the conflicting results between publications.

Since publication of the Hoi *et al. Cell* paper we have optimised the protein loading and buffers used in SPR experiments with the NTD of ClpC1 (Hawkins *et al. ACS Infect. Dis.* 2025, 3298). It is known that the protein loading (NTA chip with a His-tagged protein in the Hoi paper *vs* immobilisation on a CM5 chip in our work) can influence binding SPR (Capelli *et al. Trends Anal. Chem.* 2023, 117079). We believe our optimised SPR conditions are the most appropriate

for measuring affinities of the NRP natural products and their analogues. As suggested by the reviewer, we have added additional information into the revised manuscript to clarify this for the reader (page 10, line 272). We also note that our decision to model 2:1 binding stoichiometry for the SPR analysis was based on crystallographic data showing 2:1 binding for ecumicin (Wolf *et al. Acta Crystallogr. D Struct. Biol.* 2020, 76:458–471). That being said, the K_D for 2:1 binding ($K_{D1} = 0.39 \mu\text{M}$) and for 1:1 binding ($K_D = 0.39 \mu\text{M}$) were identical. We have added this information into the revised manuscript and have also added the sensorgram for the 1:1 binding into the revised Supplementary Information file for completeness.

Page 3

The authors describe the ClpC1P2P1 system but rely on general references that are not specific to this complex. In addition, several recent structural studies on the ClpC1P2P1 system are omitted – including structures of ClpP1P2, ClpC1, and two recent publications on the ClpC1P2P1 complex. These should be properly cited and discussed.

Please verify whether reference 15 is correct and appropriate in the context where ClpP1 and ClpP2 association for protein hydrolysis is discussed. It would be more relevant to cite the ClpP1P2 structural studies from the Sauer and Goldberg groups.

We thank the reviewers for pointing this out. We have now cited recent structural studies on Mtb ClpP1P2, ClpC1, and the ClpC1P2P1 complex (Weinhäupl, Fraga and coworkers (2025) bioRxiv and *J. Biol. Chem.* 2022, 102553). At the reviewer's request we have included publications from the Sauer and Goldberg groups (Schmitz *et al.* PNAS 2014, 111, E4587, Li *et al. J. Biol. Chem.* 2016, 7465) when discussing ClpP1 and ClpP2 association and hydrolysis.

Page 4, line 76

The authors should mention lassomycin, as it is another well-characterized ClpC1-NTD binding natural product.

During our initial revision we opted to remove lassomycin to focus on the non-ribosomal peptide (NRP) molecules that were the focus of the study. Nonetheless, we agree that it is an important member of the ClpC1-targeting natural product family and so have reinstated lassomycin back into the manuscript on page 4 line 76.

Page 5 line 124

dCym was not first described by Hoy *et al.* – but by Barbie, P. and Kazmaier in 2016.

We have corrected the reference accordingly on page 5 line 125

Mechanistic discussion

There are fundamental differences in the *in vitro* mechanisms of ecumicin, cyclomarin, and rufomycin when tested with purified proteins. Notably, cyclomarin has not been described as a strong ClpC1P2P1 inhibitor; in fact, prior studies suggest that it activates the ClpC1P1P2 complex (see Mogk Heidelberg group on cyclomarin publications). This distinction should be discussed.

Thank you for raising this important point. The literature on cyclomarin is indeed complex, with some studies showing activation of the ClpC1P1P2 complex *in vitro* rather than inhibition. However, direct comparisons between studies are challenging due to differences in assay design, substrates, ClpC constructs, compound, and compound concentrations, which vary between our study and those by the Mogk group (Maurer *et al. Cell Chem. Biol.* 2019, 26: 1169-1179, Taylor *et al. J. Biol. Chem.* 2022, 102202). We have included this in our discussion in the revised manuscript on page 13, line 396.

Page 8, line 209

The statement that IDPs are structurally similar to unfolded substrates is inaccurate. IDPs are typically stable and functional, whereas unfolded substrates are often prone to aggregation. This section should be clarified to avoid conceptual confusion.

Thank you for highlighting this distinction. We have revised the sentence to accurately reflect the differences between intrinsically disordered proteins (IDPs) and unfolded substrates (page 8, line 210).

Page 8, line 232

The manuscript states that bortezomib has no effect on *in vitro* degradation of casein by the ClpC1P2P1 complex, using a concentration of 100 μM . This seems inconsistent and potentially confusing. 100 μM is an extremely high concentration; please justify this choice and, by the way, the reference cited shows that bortezomib activates protein degradation at lower concentrations than 100 μM . Of note, the effect of the active site activators is also well described in the literature as a bell-shaped curve – that is, activation at low concentrations but inhibition at higher concentrations – please use the citations in the proper context.

In addition, why was 100 μM used *in vitro* while 5 μM ? MIC50 or MIC90 was used *in vitro*? Normally, *in vitro* assays require lower concentrations to achieve target engagement. Related to this, please clarify what were the concentrations used in the proteomics experiments? Why put the bortezomib MIC50 in the figure when you used the MIC90 in the experiment (was 100 μM the MIC90?)

Additionally, how were the *in vitro* degradation experiments performed without a ClpP1P2 activator? You observed activity in the absence of any ClpP1P2 activator? It has been reported by several groups that the ClpP1P2 is not active in the absence of active site activators. Can you explain this?

We acknowledge that 100 μM BTZ is a high concentration and have noted this in the revised manuscript (page 8, line 234). This choice was based on our standard screening approach and the expectation that higher concentrations might reveal inhibitory effects, as noted regarding bell-shaped activator profiles. The Zhou *et al. (Nat. Commun.* 2025, 16:3466) study was published after our experimental design. However, their data show only mild activation (10–20%) of FITC-casein degradation at 80–160 μM . Given our endpoint-based assay lacks continuous monitoring and fluorescent reporters, modest activation (expected 10–20%) would likely remain undetected. Therefore, we consider that our data are consistent with the cited study.

Concentrations of all compounds used in proteomic experiments have now been reported more explicitly in the revised manuscript (page 21, line 673). MIC₅₀ and MIC₉₀ values for all compounds were reported in the Supplementary Information file.

Regarding the requirement for ClpP1P2 activators, while early studies suggested that synthetic N-blocked dipeptide activators (e.g., Z-Leu-Leu) were necessary for ClpP1P2 activity, subsequent work has demonstrated that this is not the case when AAA⁺ partners and substrates are present (Leodolter *et al.* PloS One 2025, 10:e0125345, Schmitz and Sauer Mol. Microbiol. 2014, 93:617-628). Consistent with this, studies from our laboratories have routinely performed degradation assays without activators and observed robust activity (Hoi *et al.* Cell 2023, 186: 2176–2192; Morreale *et al.* Cell 2022, 185: 2338–2353).

Page 10, line 281

The authors state that Hsp20 is specifically enriched by NRP action, yet a similar enrichment is observed with bortezomib – a compound that does not bind to ClpC1 and preferentially targets the M. tuberculosis proteasome rather than ClpP1P2. This raises concerns about the specificity of the reported effect. The data appear to contradict the claim of selective enrichment – likely any stress in protein quality control will lead to an Hsp20 increase. Of note, the authors choose not to test the binding of the compounds to other chaperones as we suggested before, we do understand that – but we do believe you would likely get binding as well – but we agree this is not a key issue for review.

We have clarified this in the discussion (page 10, line 288) and avoided referring to this effect as selective. While BTZ also induces Hsp20, it does so as part of a broad PQC response involving all major chaperones and proteases. In contrast, NRPs trigger what we had previously coined selective changes, with convergence only at Hsp20 for Ecu* and IlaE; while dCym does not induce Hsp20. Thus, Hsp20 upregulation is not universal and its strong induction by Ecu*—comparable to BTZ despite the absence of a broader PQC response—remains noteworthy.

Regarding the reviewer’s comment on the potential interaction of NRPs with other chaperones, we agree that it represents an exciting avenue for future investigation, and we have mentioned this in the discussion (page 15, Line 454).

Heat stress section

The discussion of heat stress is unclear. Are the authors suggesting that heat stress potentiates the effects of drugs acting on protein quality control? If so, this concept is already well established in the literature and does not constitute a novel finding. The claims in this section should be moderated accordingly, as the data is largely confirmatory.

We have moderated the language in this section to clarify that our findings are confirmatory and consistent with prior literature. (Page 12, Line 348, Page 12, Line 363).

Reviewer #2 (Remarks to the Author):

Reviewer #3 (Remarks to the Author):

The authors have done an excellent job responding to the prior critique. No further changes are suggested.

We thank the reviewer for their positive feedback.

Reviewer #4 (Remarks to the Author):

The authors have addressed my previous comments and have now submitted an improved version of the manuscript.

We thank the reviewer for their positive feedback.